# Dissecting the mechanism of atlastin-mediated homotypic membrane fusion at the single-molecule level

Lijun Shi[1,7], Chenguang Yang [2,3,7], Mingyuan Zhang[4], Kangning Li[1], Keying Wang[4], Li Jiao[5], Ruming Liu[5], Yunyun Wang[1], Ming Li [2], Yong Wang [4,6] ✉, Lu Ma[2] ✉, Shuxin Hu[2] ✉ & Xin Bian [1] ✉

Homotypic membrane fusion of the endoplasmic reticulum (ER) is mediated by dynamin-like GTPase atlastin (ATL). This fundamental process relies on GTP-dependent domain rearrangements in the N-terminal region of ATL (ATL$_{cyto}$), including the GTPase domain and three-helix bundle (3HB). However, its conformational dynamics during the GTPase cycle remain elusive. Here, we combine single-molecule FRET imaging and molecular dynamics simulations to address this conundrum. Different from the prevailing model, ATL$_{cyto}$ can form a loose crossover dimer upon GTP binding, which is tightened by GTP hydrolysis for membrane fusion. Furthermore, the α-helical motif between the 3HB and transmembrane domain, which is embedded in the surface of the lipid bilayer and self-associates in the crossover dimer, is required for ATL function. To recycle the proteins, Pi release, which disassembles the dimer, activates frequent relative movements between the GTPase domain and 3HB, and subsequent GDP dissociation alters the conformational preference of the ATL$_{cyto}$ monomer for entering the next reaction cycle. Finally, we found that two disease-causing mutations affect human ATL1 activity by destabilizing GTP binding-induced loose crossover dimer formation and the membrane-embedded helix, respectively. These results provide insights into ATL-mediated homotypic membrane fusion and the pathological mechanisms of related disease.

In eukaryotes, the endoplasmic reticulum (ER) spreads throughout the cell as a dynamic and continuous network of sheets and tubules with three-way junctions. This unique structure is formed by homotypic fusion between ER membranes[1,2], which is catalyzed by dynamin-like integral membrane protein atlastin (ATL) in metazoans, Sey1p in yeast, and Root Hair Defective 3 (RHD3) in plants[3-12]. Because the ER network is critical to cellular function, the specific isoforms of ATLs are associated with a number of cellular processes, including the bone morphogenetic protein signaling pathway[13-16], lipid droplet size regulation[17], nuclear envelope assembly[18], ER-phagy[19,20], COPII formation[21], autophagosome formation[22], and flavivirus replication[23,24]. Mutations in human ATL1 are implicated in hereditary spastic

[1]State Key Laboratory of Medicinal Chemical Biology, College of Life Sciences, Frontiers Science Center for Cell Responses, Nankai University, Tianjin 300071, China. [2]National Laboratory for Condensed Matter Physics, Institute of Physics, Chinese Academy of Sciences, Beijing 100190, China. [3]University of Chinese Academy of Sciences, Beijing 100049, China. [4]College of Life Sciences, Zhejiang University, Hangzhou 310027, China. [5]College of Life Sciences, Nankai University, Tianjin 300071, China. [6]The Provincial International Science and Technology Cooperation Base on Engineering Biology, International Campus of Zhejiang University, Haining 314400, China. [7]These authors contributed equally: Lijun Shi, Chenguang Yang. ✉e-mail: yongwang_isb@zju.edu.cn; luma@iphy.ac.cn; hushuxin@iphy.ac.cn; xin.bian@nankai.edu.cn

paraplegia (HSP) and hereditary sensory neuropathy type I (HSN1)[25,26]. The observation of abnormally long, non-branched ER tubules in mammalian cells expressing ATL1 with disease mutations suggests that the disease is caused by defects in ER homotypic fusion[6,10,27,28].

The membrane fusion events within a cell are classified into two types: heterotypic and homotypic fusion. Heterotypic membrane fusion has been extensively studied and shown to be driven by the folding energy from conformational changes in SNARE complexes, which are disassembled by NSF and SNAPs for recycling after fusion[29]. However, the detailed processes underlying homotypic membrane fusion mediated by ATL or other related dynamin-like proteins, such as mitochondrial outer membrane fusion protein mitofusin[30], remain elusive.

ATL comprises an N-terminal cytosolic region, including a GTPase domain and a middle domain, two transmembrane domains (TMDs), and a C-terminal tail. The crystal structures of the N-terminal cytosolic region of ATL (ATL$_{cyto}$) and its ortholog in yeast have been determined for different nucleotide-bound states[31-36]. Two different conformations of human ATL1$_{cyto}$ have been observed in the presence of GDP[31,33]. Though biochemical experiments have demonstrated that GDP-bound ATL1 is a monomer or weak dimer, both of the conformations revealed

that a crystallographic dimer formed between the two GTPase domains. Moreover, in one conformation, the middle domains that fold into three-helix bundles (3HBs) point in opposite directions (Fig. 1a, Form 1 extended dimer), whereas, in the other, the linker regions between the GTPase domains and 3HBs cross one another and the 3HBs are in a parallel orientation (Fig. 1a, Form 2 loose crossover dimer). Interestingly, these two conformations partly use the same residues involved in the interface between the GTPase domain and 3HB, though it is an intramolecular interaction in Form 1 dimer and intermolecular interaction in Form 2 crossover dimer[31]. A third conformation of ATL1$_{cyto}$ dimer (Fig. 1a, Form 3 tight crossover dimer) was obtained in the presence of non-hydrolyzable GTP analog GMppNHp or the presence of GDP/AlF$_4^-$[32] that mimics the transition state of GTP hydrolysis[37,38]. Similar to Form 2, the 3HBs in Form 3 are parallel but much closer to one another. ATL-mediated membrane fusion has been suggested to be driven by the GTP-dependent movement of its 3HB relative to the GTPase domain[31,33,39-41]. However, structural and biochemical studies of human ATLs and *Drosophila* ATL (dmATL) have led to conflicting membrane fusion mechanisms.

Three different models have been proposed to understand the homotypic membrane fusion process. In one model, ATL molecules

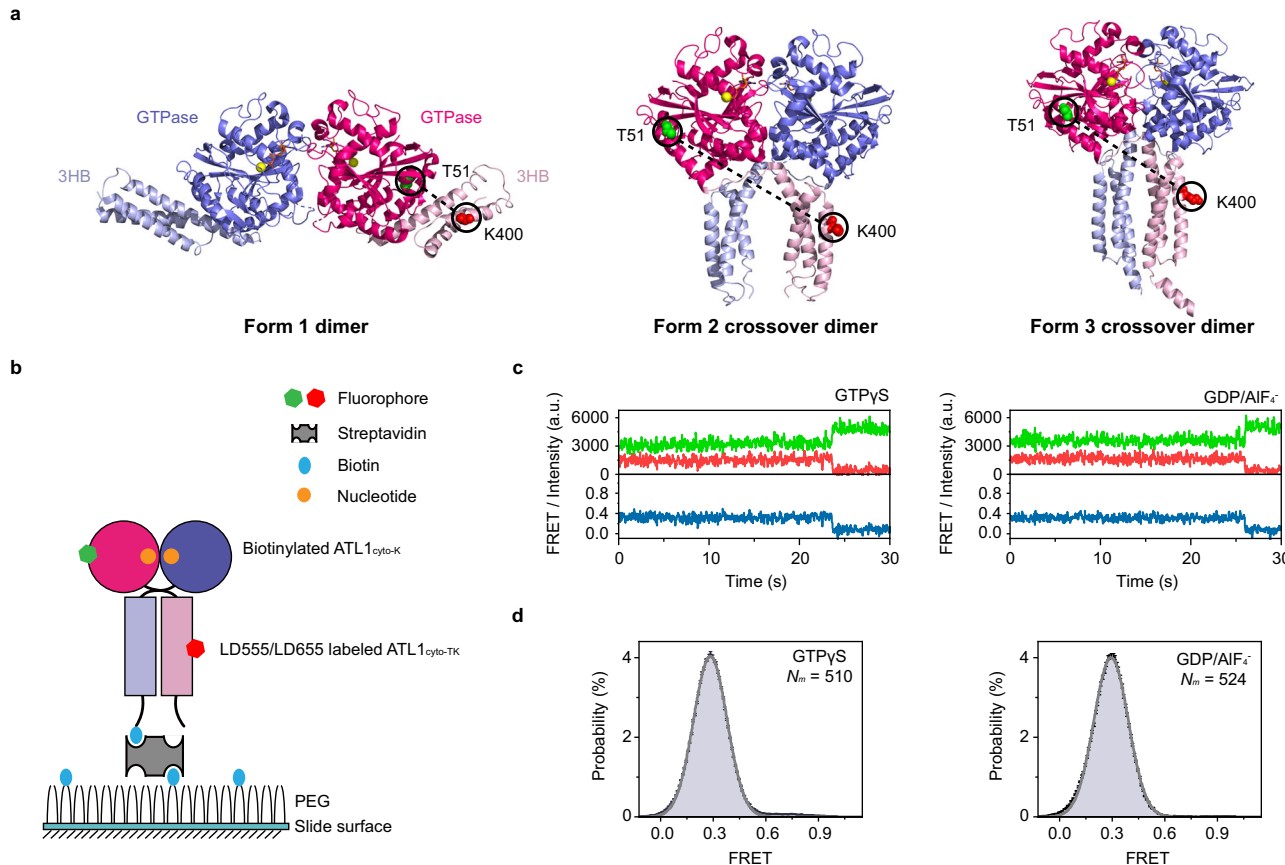

**Fig. 1 | Conformations of ATL1$_{cyto}$ dimers in the presence of GTPγS and GDP/AlF$_4^-$ revealed by intramolecular smFRET assays. a** Ribbon representations of the crystal structures of ATL1$_{cyto}$ dimers in Form 1 (left, PDB code 3QOF), Form 2 (middle, PDB code 3QNU), and Form 3 (right, PDB code 4IDN) conformations rendered in PyMOL. One protomer is shown in pink and the other in blue. The GTPase domains are shown in regular colors and the 3HBs in pale colors. The nucleotides are shown as orange sticks and the magnesium ions as yellow spheres. The selected T51/K400 pairs in one protomer for dye labeling are represented as green and red spheres, respectively. **b** Strategy of the intramolecular smFRET assays for ATL1$_{cyto-K}$-ATL1$_{cyto-TK}$ dimers in the presence of GTPγS and GDP/AlF$_4^-$. The dimer is formed by a C-terminal biotinylated ATL1$_{cyto-K}$ and LD555/LD655-labeled ATL1$_{cyto-TK}$. The biotin-streptavidin interaction was used to immobilize the

protein in a streptavidin-coated microfluidic chamber. **c** Representative fluorescence and smFRET trajectories of intramolecular smFRET assays for ATL1$_{cyto-K}$-ATL1$_{cyto-TK}$ dimers in the presence of GTPγS (left) and GDP/AlF$_4^-$ (right). LD555 (the donor) is shown in green, LD655 (the acceptor) in red, and FRET in dark blue. **d** Intramolecular smFRET distributions of ATL1$_{cyto-K}$-ATL1$_{cyto-TK}$ dimers in the presence of GTPγS (left) and GDP/AlF$_4^-$ (right). All of the individual FRET values with the number of molecules (Nm) displayed were compiled into a conformation-population histogram (gray lines) and fitted into a one-state GaussAmp distribution (~0.28). Each bar height represents the normalized count (%). The length of the error bar represents the normalized SD of a Poissom distribution from the count. Source data are provided as a Source Data file.

bind GTP and generate Form 1 dimers across different membranes, which is followed by a large conformational change from Form 1 to Form 3, triggered by GTP hydrolysis to pull the two membranes towards each other[42]. The second model, however, suggests that GTP binding is enough to induce the tight Form 3 crossover dimer formation for fusion and GTP hydrolysis is only to initiate dimer dissociation for a new round of fusion[43]. In the third model, GTP hydrolysis is proposed to occur within ATL monomers[32,44], but this model is not supported by biochemical analyses of ATL[45] and contradicts the results from protein-induced fluorescence enhancement (PIFE) experiments[43]. In addition to the controversy regarding domain rearrangements upon GTP binding and GTP hydrolysis, the ATL monomer conformations regulated by the sequential release of Pi and GDP[46] to start a new round of fusion are still poorly understood. These conundrums are due to a lack of an approach to comprehensively analyze the structural dynamics of ATL in each step of the GTPase cycle.

Single-molecule techniques have provided a unique opportunity to dissect the fundamental mechanisms of protein function by directly observing and accurately detecting the behaviors of individual molecules. Here, we develop three different experimental systems to apply intramolecular and intermolecular single-molecule Förster resonance energy transfer (smFRET) imaging to observe the conformational dynamics and pathogenic mutations of individual ATL1$_{cyto}$ molecules in solution in the absence or presence of different nucleotides. Our data provide evidence that ATL1$_{cyto}$ can adopt a loose (Form 2-like) crossover dimer conformation upon GTP binding, a surprising observation given that the current models in the literature only predict a Form 1 or Form 3 dimer conformation for GTP-bound ATL1$_{cyto}$[42,43]. The loose association of 3HBs is tightened in the transition state of GTP hydrolysis. The physiological importance of GTP hydrolysis-induced tightening of the 3HB-3HB interface is supported by the single-molecule analysis of ATL1$_{cyto}$ with a disease mutation. In addition, based on the results of single-molecule FRET imaging and molecular dynamics (MD) simulations, we found that the α-helical motif between the 3HB and TMD, which is embedded in the surface of the lipid bilayer and self-associates in the ATL dimer, is required for membrane fusion. Finally, we show that the sequential release of Pi and GDP regulates the protein recycling process in a stepwise manner. Pi release, which disassembles the dimer, activates flexibility in the GTPase domain-3HB region with a preference for the Form 1 conformation, and the subsequent GDP dissociation alters the conformational preference of monomeric ATL1$_{cyto}$ towards the Form 2 conformation to enter the next GTPase cycle. These results provide insights into the mechanisms of dynamin-like protein-mediated homotypic membrane fusion and pathological mechanisms in HSP.

## Results

### Dimerization of ATL1$_{cyto}$ induced by GTP binding and hydrolysis

The domain rearrangements of individual ATL1$_{cyto}$ molecules in solution in the absence or presence of different nucleotides were monitored using smFRET. We generated three constructs of ATL1$_{cyto}$ (residues 1–447) in which all native cysteines were substituted with alanines and T51, K400, or the T51/K400 pair was mutated to cysteine [termed Cysless ATL1$_{cyto}$-T51C (ATL1$_{cyto-T}$), Cysless ATL1$_{cyto}$-K400C (ATL1$_{cyto-K}$), and Cysless ATL1$_{cyto}$-T51C/K400C (ATL1$_{cyto-TK}$), Supplementary Figs. 1–3]. K400 was previously used as a labeling site for bulk FRET assays[44]. We also added an Avi tag followed by a His tag to the carboxyl terminus of ATL1$_{cyto-K}$ and a C-terminal His tag to ATL1$_{cyto-T}$ and ATL1$_{cyto-TK}$ (Supplementary Fig. 2 and 3). ATL1$_{cyto-TK}$ had similar GTPase activities as wild-type (WT) ATL1$_{cyto}$ (Supplementary Fig. 4). As the GTPase activity of ATL relies on intramolecular and intermolecular domain interactions[31–33,39,40], these results indicate that the mutations we introduced do not affect the GTP-dependent conformational dynamics of the protein.

We first performed photobleaching experiments using LD655-labeled ATL1$_{cyto-K}$ to test the dimerization state of the protein at the single-molecule level. The C-terminal Avi tag of LD655-ATL1$_{cyto-K}$ was biotinylated (Supplementary Fig. 5). The modified proteins were mixed with or without 1 mM GDP, GTPγS, GDP/AlF$_4^-$ at a high protein concentration of ~2 μM, followed by dilution and immobilization of the protein (~0.3 nM) in a streptavidin-coated microfluidic chamber (Supplementary Fig. 6a, b). The samples were imaged using total internal reflection fluorescence (TIRF) microscopy. At this low protein concentration, the individual fluorescently labeled particles could be distinguished (Supplementary Fig. 6c), and almost all of them displayed one-step photobleaching in the absence or presence of GDP (Supplementary Fig. 6d, f). With GTPγS or GDP/AlF$_4^-$, the fluorescence of most LD655-ATL1$_{cyto-K}$ molecules was photobleached in two steps (Supplementary Fig. 6c, e, f). These data show that the ATL1$_{cyto}$ molecules are monomeric in the absence or presence of GDP and form dimers in the presence of GTPγS or GDP/AlF$_4^-$ under the conditions of single-molecule experiments, consistent with previous bulk FRET experiments[44,45].

To assess the conformation of a protomer in the ATL1$_{cyto}$ dimer upon GTP binding or in the transition state of GTP hydrolysis using intramolecular smFRET, we labeled ATL1$_{cyto-TK}$ with the LD555/LD655 fluorophore pair[47] (Fig. 1a) and biotinylated the C-terminal Avi-tagged ATL1$_{cyto-K}$ (Supplementary Fig. 5). These two modified proteins were incubated together with 1 mM GTPγS or GDP/AlF$_4^-$ to form ATL1$_{cyto-TK}$-ATL1$_{cyto-K}$ dimer, which was subsequently diluted and immobilized in a streptavidin-coated microfluidic chamber through biotin-streptavidin binding (Fig. 1b). In this system, only the signals of fluorescently labeled ATL1$_{cyto-TK}$ that dimerized with biotinylated ATL1$_{cyto-K}$-Avi in a nucleotide-dependent manner were recorded. The corresponding FRET efficiencies for individual LD555/LD655-labeled ATL1$_{cyto-TK}$ molecules were recorded at an exposure time of 30 ms, and hundreds of smFRET trajectories exhibiting anti-correlated donor and acceptor fluorescence intensities were extracted (Fig. 1c and Supplementary Fig. 7).

According to the reported crystal structures of ATL1$_{cyto}$[31–33], labeled residues 51 and 400 in one ATL$_{cyto-TK}$ molecule are close in Form 1 dimer and distant in Form 2 and Form 3 crossover dimers (Fig. 1a). Analyses of intramolecular smFRET data showed that ATL1$_{cyto-TK}$ exhibited single low-FRET distributions centered at ~0.28 in the presence of GTPγS or GDP/AlF$_4^-$ (Fig. 1d and Supplementary Table 1). In addition, the FRET values over time for almost all ATL1$_{cyto-TK}$ molecules were stationary (Supplementary Fig. 7). A possible conformation corresponding to this low-FRET state is the Form 3 tight crossover dimer (Supplementary Tables 1 and 2), which was obtained in the GMppNHp- or GDP/AlF$_4^-$-bound condition[32]. Notably, this low-FRET state is also likely to represent a conformation similar to Form 2 crossover dimer in which the 3HBs loosely associate with one another, as the intramolecular smFRET imaging may not be able to recognize the difference between the Form 2 and Form 3 conformations. Nevertheless, we barely observed a higher FRET state corresponding to the Form 1 conformation.

Next, we performed intermolecular smFRET imaging to confirm the crossover dimer conformation in the presence of GTPγS or GDP/AlF$_4^-$. A mixture of LD555-ATL1$_{cyto-T}$ and biotinylated LD655-ATL1$_{cyto-K}$ with GTPγS or GDP/AlF$_4^-$ was diluted and added to a streptavidin-coated microfluidic chamber (Supplementary Fig. 8a, b). The immobilized individual nucleotide-dependent ATL1$_{cyto-T}$-ATL1$_{cyto-K}$ dimers were imaged over time by TIRF microscopy (Supplementary Fig. 8b). Consistent with the short distance between the two labeled sites in the Form 2 and Form 3 crossover dimers (Supplementary Fig. 8a and Table 2), the labeled ATL1$_{cyto-T}$-ATL1$_{cyto-K}$ dimer only had a single high-FRET peak centered at ~0.80 in the presence of GTPγS or GDP/AlF$_4^-$ (Supplementary Fig. 8c, d and Table 3). Thus, instead of generating a

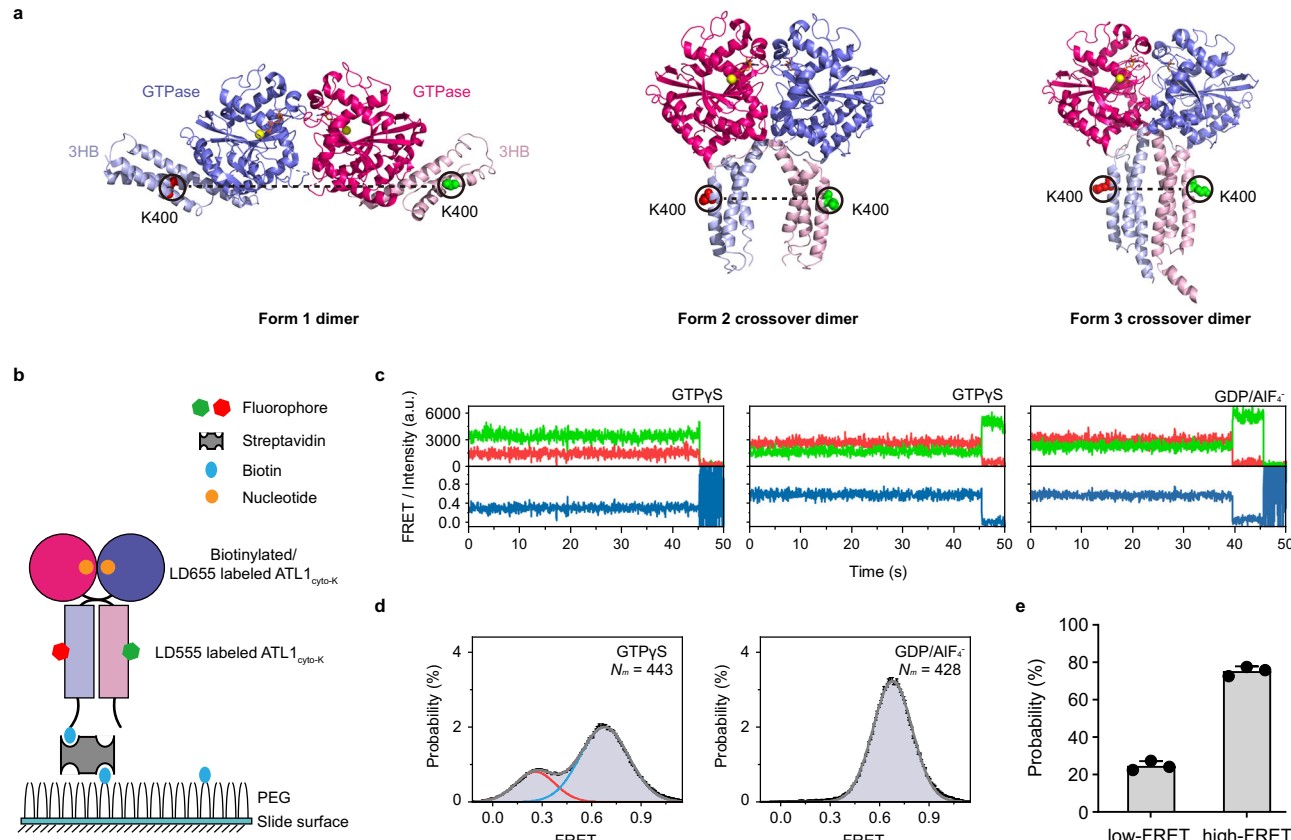

**Fig. 2 | Conformations of ATL1$_{cyto}$ dimers in the presence of GTPγS and GDP/AlF$_4^-$ revealed by intermolecular smFRET assays. a** As in Fig. 1a, but K400 in each protomer was selected for dye labeling and is represented as green and red spheres. **b** Strategy of the intermolecular smFRET assays for ATL1$_{cyto-K}$ dimers in the presence of GTPγS and GDP/AlF$_4^-$. The dimer is formed by a LD555-labeled ATL1$_{cyto-K}$ and LD655-labeled ATL1$_{cyto-K}$. One protomer is biotinylated at the C-terminus. The biotin-streptavidin interaction was used to immobilize the protein in a streptavidin-coated microfluidic chamber. **c** Representative fluorescence and smFRET trajectories of intermolecular smFRET assays for ATL1$_{cyto-K}$ dimers in the presence of GTPγS (left and middle) and GDP/AlF$_4^-$ (right). LD555 (the donor) is shown in green, LD655 (the acceptor) in red, and FRET in dark blue. **d** Intermolecular smFRET distributions of ATL1$_{cyto-K}$ dimers in the presence of GTPγS (left) and GDP/AlF$_4^-$

(right). All of the individual FRET values with the number of molecules (Nm) displayed were compiled into a conformation-population FRET histogram (gray lines) and fitted into a two-state GaussAmp distribution (left, centered at ~0.28 in red line and ~0.66 in blue line) and one-state GaussAmp distribution (right, centered at ~0.66). Each bar height represents the normalized count (%). The length of the error bar represents the normalized SD of a Poisson distribution from the count. **e** The relative occupancy of the low-FRET state and high-FRET state for ATL1$_{cyto-K}$ dimers in the presence of GTPγS as derived from histograms. Data are presented as mean ± SD (*n* = 3 independent experiments) determined from three randomly assigned populations of all FRET traces. Source data are provided as a Source Data file.

Form 1 extended dimer, GTP binding to ATL1$_{cyto}$ induces crossover dimer formation.

## GTP binding generates loose ATL1$_{cyto}$ crossover dimer, which is tightened by GTP hydrolysis

To further explore the conformational preferences of ATL1$_{cyto}$ dimer in the presence of GTPγS and GDP/AlF$_4^-$, we mixed LD555-ATL1$_{cyto-K}$ and biotinylated LD655-ATL1$_{cyto-K}$ with 1 mM GTPγS or GDP/AlF$_4^-$ and monitored the corresponding FRET values for the immobilized individual nucleotide-dependent ATL1$_{cyto-K}$ dimers over time (Fig. 2a, b). Surprisingly, in the presence of GDP/AlF$_4^-$, we observed a high-FRET state corresponding to Form 3 crossover dimer conformation, whereas GTPγS-loaded ATL1$_{cyto-K}$ exhibited both a high-FRET distribution centered at ~0.66 and low-FRET peak centered at ~0.28 (Fig. 2c–e, Supplementary Fig. 9 and Table 4). Transitions between low- and high-FRET states were rare (Supplementary Fig. 9), but it is possible that the ATL1$_{cyto}$ dimers undergo a low-frequency transition between low- and high-FRET states on a sub-millisecond timescale in the presence of GTPγS, and this fast transition could not be efficiently captured by our smFRET setup (30 ms per frame). When the ATL1$_{cyto}$ molecules dimerize, the distance between labeled position 400 in two

3HBs is shortest in Form 3 (Fig. 2a and Supplementary Table 2). Given that the FRET efficiency negatively correlates with the distance between the donor and acceptor fluorophores and no Form 1 extended dimer was formed in the presence of GTPγS or GDP/AlF$_4^-$ (Fig. 1a–d and Supplementary Fig. 8), these data demonstrate that the GTP-bound ATL1$_{cyto}$ crossover dimers adopt two conformations corresponding to the loose crossover dimer and Form 3 tight crossover dimer, respectively, and the tight association of 3HBs is completely formed in the transition state of GTP hydrolysis.

## Disease mutation S398Y in ATL1 affects the formation of loose crossover dimer

HSP has been pathogenically linked to defects in ER homotypic fusion[6,10,27,28]. Most ATL1 mutations causing HSP localize at the intermolecular and intramolecular domain interfaces and exhibit weakened nucleotide binding and/or reduced propensity for dimerization[33]. However, some mutations cause disease despite exhibiting WT levels of nucleotide-dependent dimerization when examined by commonly used biochemical and biophysical methods[33]. One such mutation is S398Y, which localizes at the center of the second α-helix of 3HB (Fig. 3a and Supplementary Fig. 1).

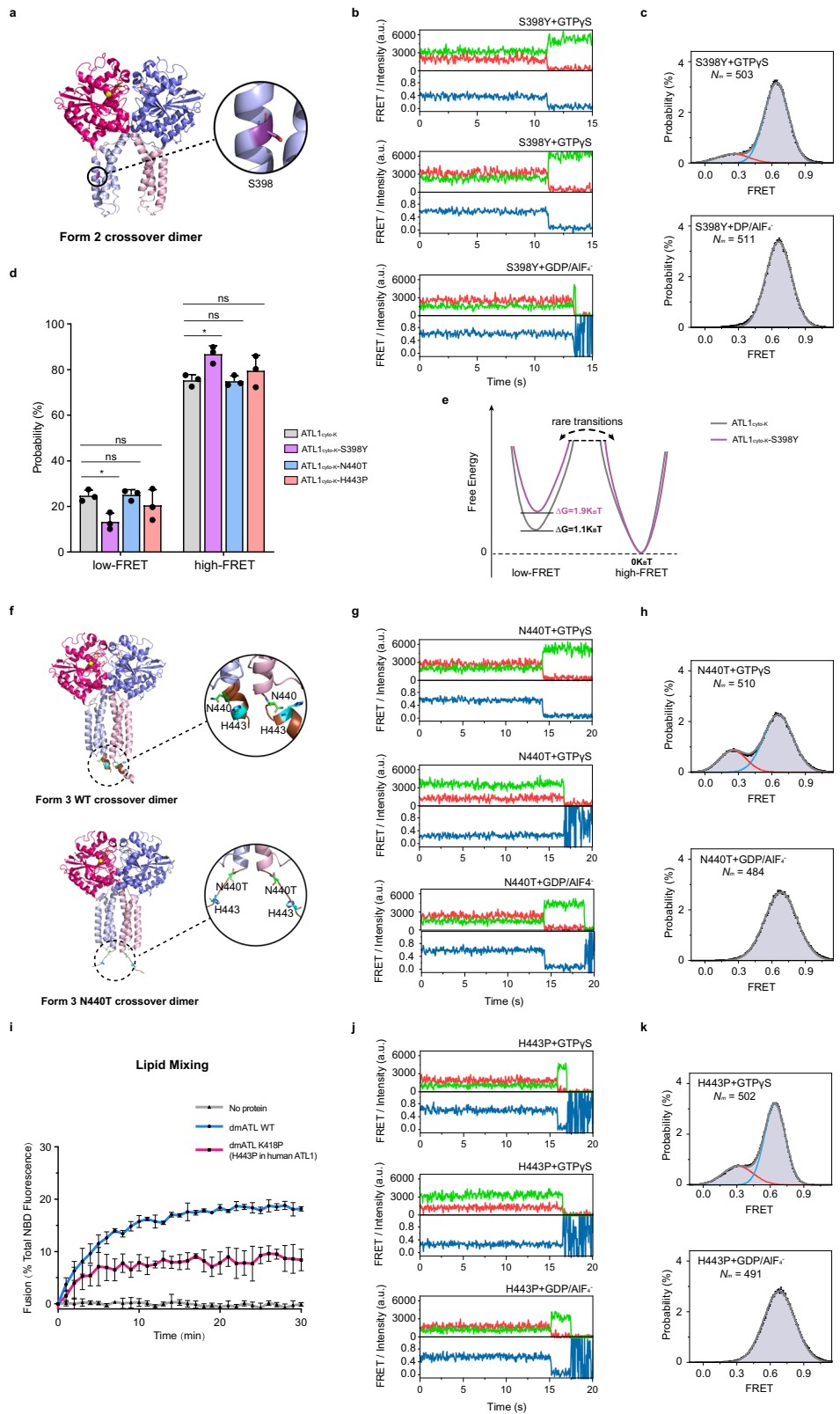

To further understand the pathological mechanisms in HSP, we tested the activity of ATL1 with a disease mutation in vivo. Yeast cells lacking Sey1p and another ER-shaping protein, Yop1p (DKO), had a significant reduction in the intricate tubular ER network at the periphery of the cell due to a defect in ER membrane fusion, and the expression of WT ATL1 can rescue this abnormal cortical ER morphology (Supplementary Fig. 10)[3,6,48]. However, in the DKO cells expressing disease mutant ATL1 S398Y, restoration of the tubular ER network to the WT levels was not observed (Supplementary Fig. 10). Next, we applied intermolecular smFRET experiments to the mutant ATL1$_{cyto-K}$ carrying S398Y (Fig. 2b). In the presence of GDP/AlF$_4^-$, the FRET trajectories and distributions of ATL1$_{cyto-K}$-S398Y were reminiscent of ATL1$_{cyto-K}$ (Fig. 3b, c, Supplementary Fig. 11 and Table 4). However, the occupancy of the low-FRET state observed with GTPγS

**Fig. 3 | Loose crossover dimer formation and the linker region between the 3HB and TMD play roles in ATL-mediated membrane fusion. a** Ribbon representation of the crystal structure of ATL1$_{cyto}$ dimer in Form 2 (PDB code 3QNU) conformation rendered in PyMOL. The molecules are shown as in Fig. 1a. S398 is represented as a purple stick in one protomer. **b, c** As in Fig. 2c, d, but with ATL1$_{cyto-K}$-S398Y. **d** Relative occupancy of the low-FRET state and high-FRET state for ATL1$_{cyto-K}$, ATL1$_{cyto-K}$-S398Y, ATL1$_{cyto-K}$-N440T, and ATL1$_{cyto-K}$-H443P dimers in the presence of GTPγS as derived from histograms. The data from Fig. 2e for GTPγS-bound ATL1-cyto-K dimer are shown for comparison. Data are presented as mean ± SD ($n$ = 3 independent experiments), determined from three randomly assigned populations of all FRET traces. ns, not significant; *$P$ < 0.05 by one-way ANOVA with Tukey's multiple comparisons test. $P$-values: 0.0441 (ATL1$_{cyto-K}$ vs ATL1$_{cyto-K}$-S398Y), 0.9995 (ATL1$_{cyto-K}$ vs ATL1$_{cyto-K}$-N440T), 0.6430 (ATL1$_{cyto-K}$ vs ATL1$_{cyto-K}$-H443P).

**e** Relative free energy models of ATL1$_{cyto-K}$ and ATL1$_{cyto-K}$-S398Y in the presence of GTPγS. The differences between the low-FRET state and high-FRET state are roughly scaled and marked below the corresponding states. **f** Ribbon representations of the crystal structures of ATL1$_{cyto}$ (top, PDB code 4IDN) and ATL1$_{cyto}$-N440T (bottom, PDB code 4IDP) dimer in Form 3 conformation rendered in PyMOL. The molecules are shown as in Fig. 1a. The linker regions between the 3HBs and TMDs are colored in brown. The N440 (N440T) and H443 are represented as green and cyan sticks, respectively. **g, h** As in Fig. 2c, d, but with ATL1$_{cyto-K}$-N440T. **i** Time courses of membrane fusion between donor and acceptor proteoliposomes containing WT dmATL, dmATL K418P (H443P in human ATL1), or no protein in the presence of GTP as assessed by dequenching of NBD-PE fluorescence (mean ± SD, $n$ = 3 independent experiments). **j, k** As in Fig. 2c, d, but with ATL1$_{cyto-K}$-H443P. Source data are provided as a Source Data file.

was decreased in the case of ATL1$_{cyto-K}$-S398Y (Fig. 3b–d, Supplementary Fig. 11 and Table 4). This change revealed that mutating Ser to Tyr at position 398 of ATL1 destabilizes the Form 2-like crossover dimer in the presence of GTPγS with differences in the relative free energy of -0.8 $k_B$T (Fig. 3e). Analyses of this disease mutation imply that the formation of loose crossover dimer upon GTP binding is physiologically important.

### The helical motif between the 3HB and TMD is essential for membrane fusion

Another HSP-causing ATL1 mutation, N440T, localizes to the linker region between the 3HB and TMDs (Fig. 3f and Supplementary Fig. 1). When we used intermolecular smFRET to examine ATL1$_{cyto-K}$ carrying N440T in the presence of GTPγS or GDP/AlF$_4^-$ (Fig. 2b), the FRET trajectories and distributions of ATL1$_{cyto-K}$-N440T were similar to ATL1$_{cyto-K}$ (Fig. 3d, g, h, Supplementary Fig. 12, and Table 4). These results indicate that this disease mutation does not affect nucleotide-dependent dimerization of ATL. To gain insights into the mechanisms underlying disease pathogenesis, we inspected the reported crystal structures of GMppNHp-bound WT ATL1$_{cyto}$ and ATL1$_{cyto}$-N440T[32]. Interestingly, a helix formed in the linker region of WT ATL1$_{cyto}$, but it became unstructured when mutating N440 to Thr (Fig. 3f).

To confirm the critical role of the helix in the linker region in the function of ATL, we introduced a proline substitution at position 443 of ATL1 and corresponding position 418 of dmATL, which is in the middle of the linker region. This mutation is expected to disrupt the structure of the α-helix (Fig. 3f and Supplementary Fig. 1). In membrane fusion assays[9,31], we reconstituted full-length dmATL into donor and acceptor proteoliposomes. The fusion of fluorescently labeled donor proteoliposomes with unlabeled acceptor proteoliposomes resulted in dilution and dequenching of the FRET pair, NBD-PE and Rhodamine-PE, showing increased NBD fluorescence. Though robust membrane fusion was achieved by WT dmATL in the presence of GTP, dmATL-H418P had significantly lower fusion activity (Fig. 3i). In DKO yeast cells, the N440T and H443P mutants did not have fenestrated ER restored to WT levels (Supplementary Fig. 10). In agreement with the results of the disease mutation experiments, this fusion defect was not caused by an impairment in crossover dimer formation, as the GTPγS- and GDP/AlF$_4^-$-dependent intermolecular smFRET trajectories and distributions of ATL1$_{cyto-K}$-H443P were highly consistent with ATL1$_{cyto-K}$ and ATL1$_{cyto-K}$-N440T (Fig. 3d, j, k, Supplementary Fig. 13 and Table 4).

To investigate the role of the helical motif linking the 3HB and TMD in ATL function, we studied the location of the helical motif in the membrane. We modeled the full-length ATL1 protein, including the TMDs and C-terminal tail, in the Apo monomer (Fig. 4a) and GDP/Pi-bound Form 3 tight crossover dimer conformations (Fig. 4b) using AlphaFold2 prediction with the AF2complex algorithm[49]. The Form 3 conformation represents the post-fusion state of ATL1 dimer (see Discussion). We then used the OPM web server[50] to predict how these

models of ATL1 interact with the membrane. The prediction suggested that the helical motif is located at the surface of the outer leaflet of the membrane in both forms (Fig. 4a, b). We also performed all-atom MD simulations of ATL1 monomer and Form 3 crossover dimer to determine how they behave in an ER-like membrane environment (Supplementary Table 5). The simulations showed that the helical motif inserted slightly deeper into the membrane (Fig. 4c, d). In particular, the hydrophobic residues oriented their side chains towards the nonpolar phase of the lipid bilayer, and the polar residues oriented their side chains towards the solvent (Fig. 4b). This orientation may help optimize the energetics of the ATL1-membrane interaction.

Interestingly, in the Form 3 tight crossover dimer, strong interactions exist between the helical motifs from each protomer over the 1-μs MD simulation (Fig. 4b, e, Supplementary Fig. 14a and Supplementary Movie 1), resulting in extensive intermolecular contacts (Fig. 4f). Notably, we did not observe any interaction between the TM domains in the MD simulations, but this may depend on the accuracy of the AF2 prediction, which needs further investigation. We also performed all-atom MD simulations of the ATL1 dimer in Form 2 conformation and observed that the helical motifs in the dimer approached and interacted with each other during the 1-μs simulation (Supplementary Fig. 14b). However, the 3HBs did not move closer within this time scale, which confirms that the assembly of the 3HBs and the interaction between the helical motifs are independent, and this helical motif has strong potential to self-associate. To validate the experimental observations, additional simulations for the GDP/Pi-bound Form 3 dimer were initiated using the crystal structure of the ATL1$_{cyto}$-N440T[32], in which the helical motif is unstructured. We observed a significant decrease in interactions between the helical motifs (Fig. 4g, h and Supplementary Fig. 14c).

Therefore, we propose that the membrane-embedded helical motifs between 3HBs and TMDs in the ATL crossover dimer interact with each other at the initial stage of lipid mixing to allow the energy provided during the tight association of 3HBs to be transferred all the way into the membrane to complete membrane fusion.

### Conformational dynamics of ATL1$_{cyto}$ regulated by Pi and GDP release

Finally, we used intramolecular smFRET imaging to analyze the behaviors of ATL1$_{cyto-TK}$ with or without GDP (Fig. 1a), which represent the sequential release of Pi and GDP[27,44,45] for dimer disassembly and protein recycling[33,40,51]. The ATL1$_{cyto-TK}$ monomers labeled with LD555/LD655 in the absence or presence of 1 mM GDP were immobilized in a streptavidin-coated microfluidic chamber through an interaction between its C-terminal His tag and biotinylated anti-His antibody (Fig. 5a). Interestingly, the FRET distributions of individual ATL1$_{cyto-TK}$ in the absence or presence of GDP were broad with an ultra-low-FRET peak centered at -0.18, medium-FRET peak centered at -0.41, high-FRET peak centered at -0.63, and ultra-high-FRET peak centered at -0.83 (Fig. 5b, c and Supplementary Table 6).

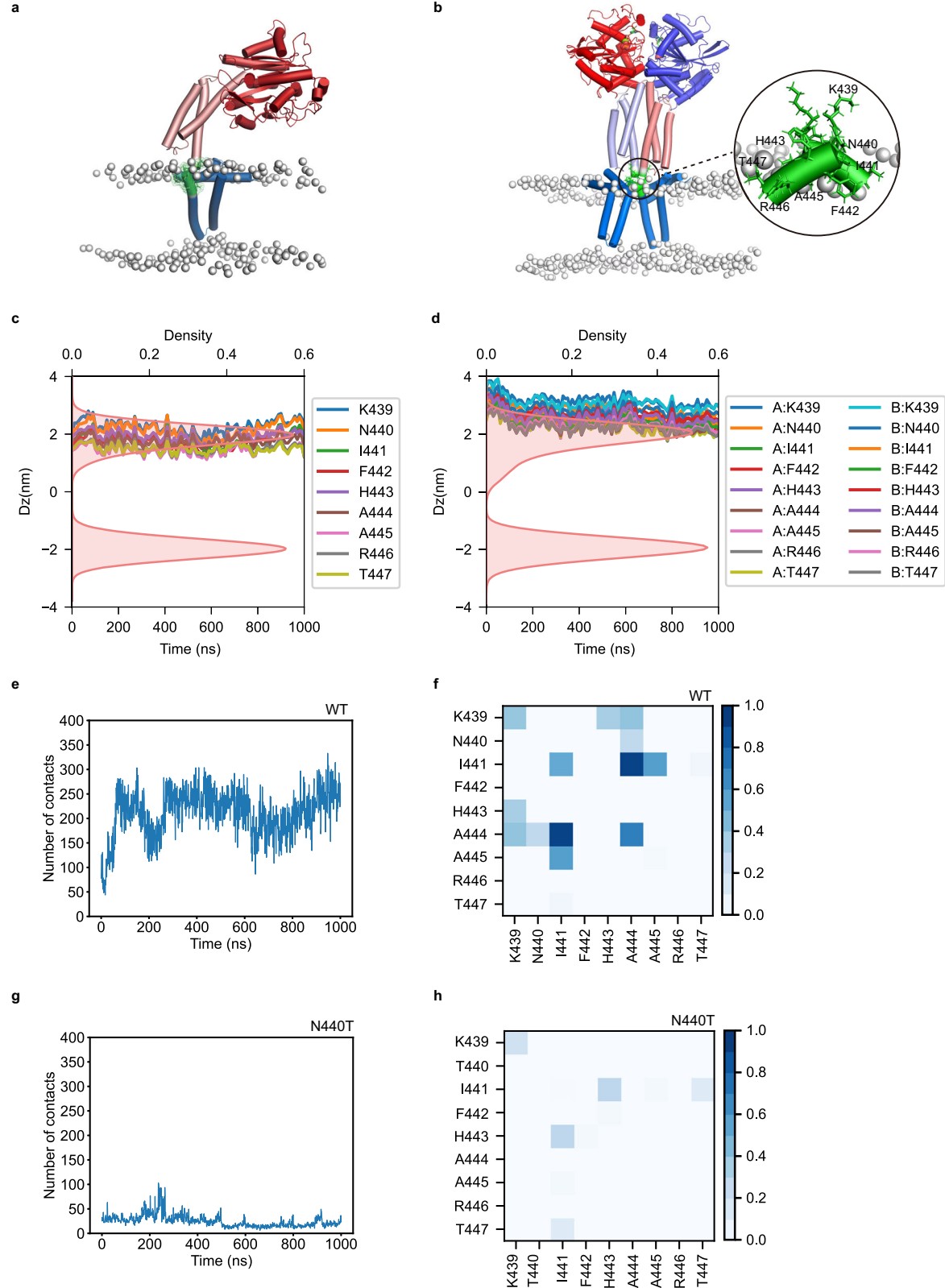

**Fig. 4 | Interaction and membrane location of the helical motifs linking the 3HBs to the TMDs of ATL1 molecules in molecular dynamics simulations.** **a**, **b** Apo ATL1 monomer (**a**) and GDP/Pi-bound ATL1 Form 3 dimer (**b**) in the membrane after 1-μs MD simulation. The helical motifs between the 3HBs and TMDs are colored in green. **c**, **d** Distance of each residue of the helical motif to the membrane center from the simulation of monomer (**c**) and GDP/Pi-bound Form 3 dimer (**d**). The red areas show the membrane thickness based on the phospholipid density. **e** MD trajectory of the number of contacts between the helical motifs in Form 3 dimer. **f**, Contact map of the helical motifs from the simulation of Form 3 dimer. **g**, **h** As in (**e**, **f**), but with ATL1-N440T.

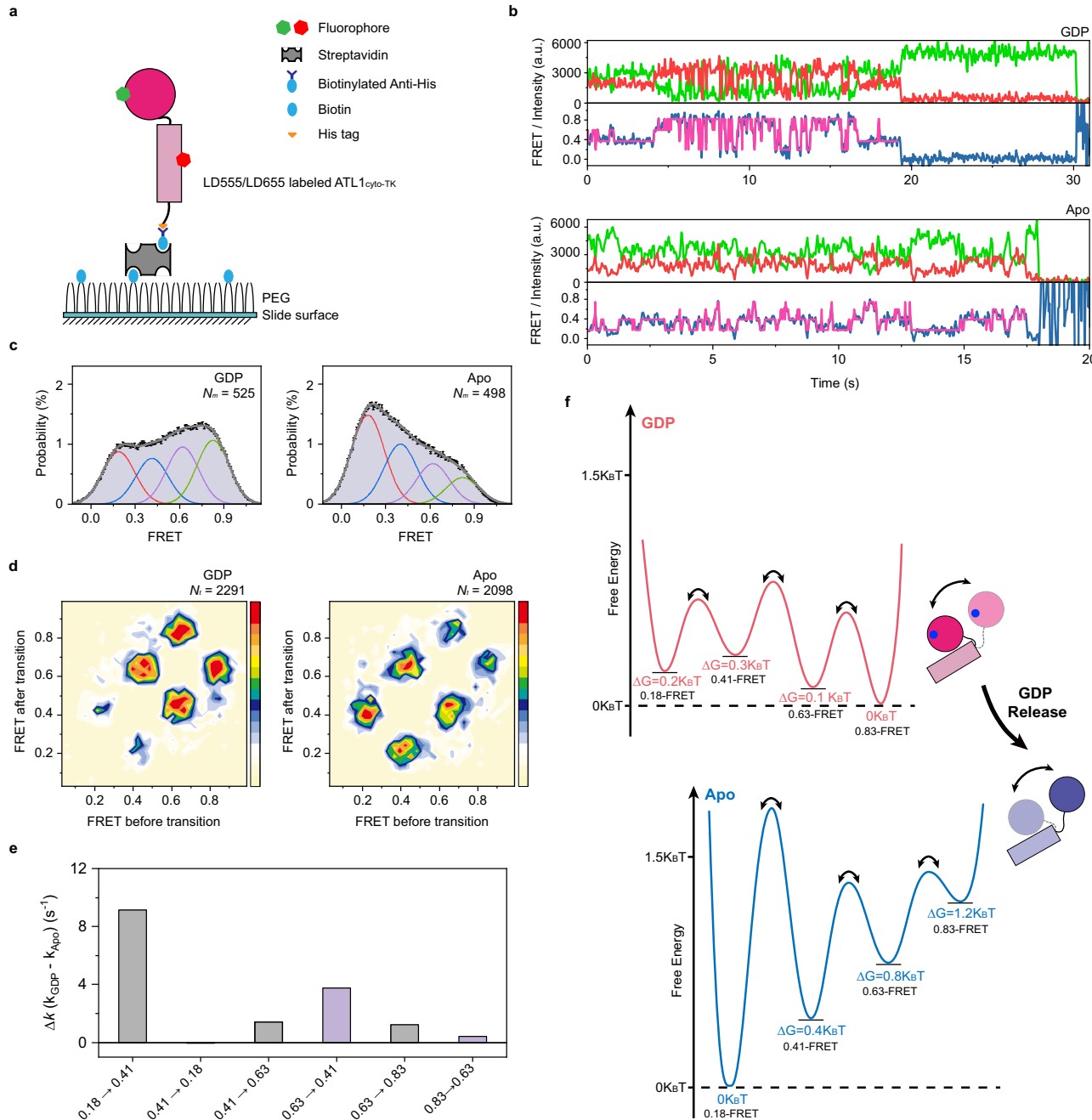

**Fig. 5 | Conformational dynamics of ATL1$_{cyto-TK}$ monomers in the absence or presence of GDP revealed by intramolecular smFRET assays. a** Strategy of the intramolecular smFRET assays for ATL1$_{cyto-TK}$ monomers in the absence or presence of GDP. The interaction between the His tag and biotinylated anti-His antibody was used to immobilize the protein in a streptavidin-coated microfluidic chamber. **b** Representative fluorescence and smFRET trajectories of intramolecular smFRET assays for ATL1$_{cyto-TK}$ monomers in the absence (bottom) or presence of GDP (top). LD555 (the donor) is shown in green, LD655 (the acceptor) in red, and FRET traces are displayed as a dark blue line for FRET efficiency and pink line for hidden Markov model initialization. **c**, Intramolecular smFRET distributions of ATL1$_{cyto-TK}$ monomers in the presence (left) or absence of GDP (right). All individual FRET values with the number of molecules (Nm) displayed were compiled into a conformation-population FRET histogram (gray lines) and fitted into a four-state GaussAmp distribution as ultra-low-FRET state (centered at -0.18), medium-FRET

state (centered at -0.41), high-FRET state (centered at -0.63), and ultra-high-FRET state (centered at -0.83) shown in red, blue, purple, and green lines, respectively. Each bar height represents the normalized count (%). The length of the error bar represents the normalized SD of a Poissom distribution from the count. **d** TDP plots under different conditions. Average FRET values before transition versus after transition are displayed as a 2D chart (scale at right) with the number of transitions (Nt) shown above. **e** Kinetic analysis of the ATL1$_{cyto-TK}$ monomers in the absence or presence of GDP. The differences in transition rate constants shown in Supplementary Fig. 15 between GDP-bound and nucleotide-free ATL1$_{cyto-TK}$ are shown. **f** Relative free energy model of the conformational dynamics of ATL1$_{cyto-TK}$ in the absence (bottom) or presence of GDP (top). The ΔG of the ultra-low-FRET state compared with the medium-FRET state, high-FRET state, and ultra-high-FRET state is roughly scaled and marked below the corresponding states. Source data are provided as a Source Data file.

To probe the kinetics of the intramolecular conformational transitions, we fitted the individual intramolecular smFRET trajectories of the ATL1$_{cyto-TK}$ monomers in the absence or presence of GDP using a four-state Hidden Markov Model process (Fig. 5b). The FRET efficiencies before and after each transition were plotted as a transition density plot (TDP) to indicate the frequencies of conformational transitions (Fig. 5d). In contrast to the GTPγS- and GDP/AlF$_4^-$-bound ATL1$_{cyto}$ dimers, which have a relatively stable GTPase domain-3HB region (Fig. 1c and Supplementary Fig. 7), frequent relative movements between the GTPase domain and 3HB of monomeric ATL1$_{cyto}$ were observed in Apo and GDP-bound conditions, reflecting a weakened interaction between the GTPase domain and 3HB (Fig. 5d). In the presence of GDP, transitions between medium- and high-FRET states and between high- and ultra-high-FRET states were predominantly observed, whereas the Apo form had frequent transitions between medium- and ultra-low-FRET states (Fig. 5d).

As expected, the conformations of Apo and GDP-bound ATL1$_{cyto-TK}$ were not identical. The ATL1$_{cyto-TK}$ monomer in the presence of GDP preferred the ultra-high-FRET Form 1 conformation (Fig. 5c, Supplementary Tables 2 and 6), whereas nucleotide-free ATL1$_{cyto-TK}$ had a dominant occupancy in the ultra-low-FRET state (Fig. 5c), which represents the Form 2 or Form 3 conformation of ATL1$_{cyto-TK}$ as a monomer (Supplementary Tables 2 and 6). This change was the result of a dramatically decreased transition rate from the ultra-low-FRET state ($\sim 0.18$) toward the medium-FRET state ($\sim 0.41$) (Fig. 5e and Supplementary Fig. 15). The energy barriers for the transition towards a high-FRET state ($\sim 0.63$) and subsequent ultra-high-FRET state ($\sim 0.83$) were also elevated. These analyses result in a qualitative model of a GDP release-altered conformational preference of ATL1$_{cyto}$ (Fig. 5f).

Collectively, these results demonstrate that the GTPase domain-3HB region of ATL becomes flexible after GTP hydrolysis and the sequential release of Pi and GDP generates two different conformational preferences of the ATL monomer in a stepwise manner to recycle the protein for a new round of fusion.

## Discussion

In this study, we measured smFRET efficiency to monitor the conformations of ATL1$_{cyto}$ and its pathogenic mutations in the absence or presence of different nucleotides, despite some discrepancies between the theoretical dye-dye distances calculated by the FRET-predict tool (Supplementary table 2)[52,53] and the experimental dye-dye distances based on the smFRET efficiency (see Methods for calculation, Supplementary tables 1, 3, 4, and 6), which could be attributed to uncertainties in the Förster radius $R_0$ and limited MD sampling. Our results partly confirm the prior mechanistic model for ATL function in which the loading of GTP induces the crossover dimerization of ATL[43]. However, our proposed model in this study also differs from previous studies. First, we show that the energy input from GTP binding is not enough to complete the tightest (Form 3) crossover dimer formation, as a proportion of ATL1$_{cyto}$ molecules in the presence of GTPγS interact as in the Form 2-like conformation, where their 3HBs loosely associate with one another (Fig. 6, Step I). Given that only the N-terminal cytosolic region of ATL1 in solution was used here, more membrane-anchored full-length ATL molecules are assumed to have a Form 2-like crossover dimer conformation upon GTP binding due to the electrostatic repulsive force between the two apposed membranes, which increases the difficulty for tight association of 3HBs. Consistent with this proposed model, GTP binding to ATL has been reported to be insufficient to induce the formation of membrane tethers[45,54]. Another important finding was that, although GTP hydrolysis did not trigger a large conformational change in ATL1$_{cyto}$ (from Form 1 to Form 3) as suggested previously[42], it did tighten the 3HB-3HB interface, allowing multiple ATL molecules on each membrane to cooperate to promote tethering and fusion between two bilayers (Fig. 6, Step II and II').

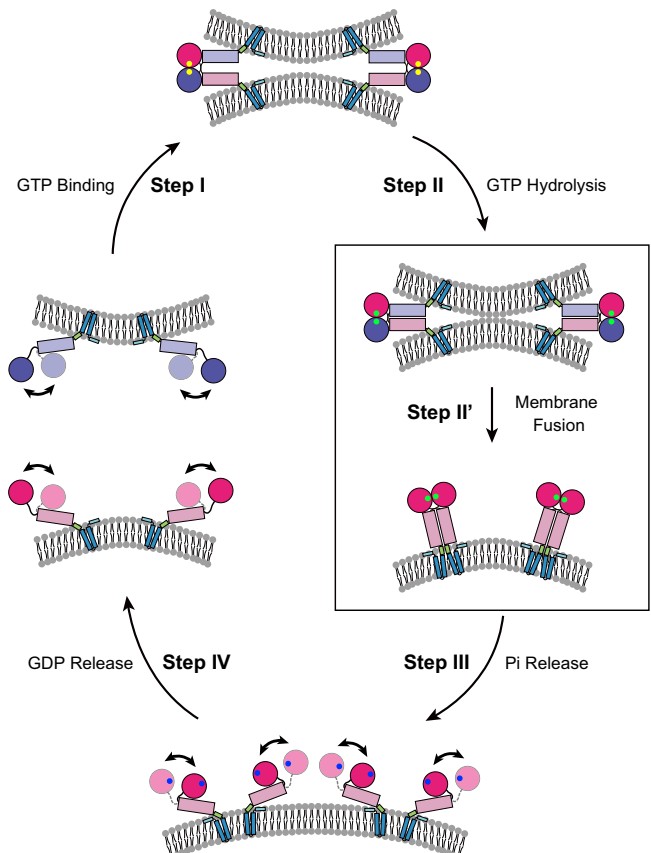

**Fig. 6 | Model of a successful membrane fusion event mediated by ATL.** Step I, monomeric ATLs sitting in two membranes generate Form 2-like loose crossover dimers upon GTP binding. Step II and II', the 3HBs of ATL dimers are tightened by GTP hydrolysis to merge the two membranes. Self-association of the membrane-embedded helix formed by the linker region between the 3HB and TMD is required to transfer the energy gained from the conformational change in ATL to the membrane to complete fusion. Step III, the release of inorganic phosphate weakens the interaction between the GTPase domain and 3HB and activates the flexibility in the GTPase domain-3HB region of ATL, which exhibits a Form 1 conformational preference for dimer disassembly. Step IV, GDP release alters the conformational preference of monomeric ATL for the Form 2 or Form 3 conformation, getting ready for the next reaction cycle. GTP, GDP/Pi, and GDP are displayed as yellow, green, and blue spheres, respectively.

After the membranes are fused by ATL, Pi and GDP are released in a sequential manner to disassemble ATL dimers for new rounds of fusion. Another unanswered question prior to this study was the domain rearrangements of ATL coupled with this process. On the basis of our smFRET results, the sequential release of Pi and GDP regulates the conformational changes in ATL in a stepwise manner for protein recycling. Unlike the membrane fission-specific dynamin superfamily GTPases, such as MxA, in which the flexibility in the GTPase domain-BSE region is released upon GTP binding[55], the frequent relative movement between the GTPase domain and 3HB of ATL was activated by Pi release with a preference for the Form 1 conformation (Fig. 6, Step III). The activation of flexibility in the GTPase domain-3HB region, which reflects the weakened interaction between the GTPase domain of one protomer and 3HB of the other promotor, is expected to initiate disassembly of the ATL dimer, and the relatively small interfacial area in the Form 1 dimer may facilitate this process. We also found that the GTPase domain-3HB region of ATL1 remains flexible in the Apo state, but GDP release leads to alternation of the conformational preference towards a Form 2 or Form 3 conformation (Fig. 6, Step IV), which may lower the difficulty of GTP binding-induced Form 2-like crossover

dimer formation between ATLs sitting in apposed membranes (Fig. 6, Step I).

Single-molecule approaches also offer a solution to better understanding the pathogenetic mechanisms that are sometimes hidden. The presence of Ser at position 398, where the corresponding residues in human ATL2/3 and other species are Ala, is a unique feature of human ATL1 (Supplementary Fig. 1). Its mutation to Tyr causes HSP but has no detectable defect in nucleotide-dependent dimerization when examined by commonly used biochemical and biophysical methods[33], though the corresponding mutation in dmATL results in impaired dimer formation[27]. Via a combination of smFRET analyses and imaging experiments using yeast cells, our results indicated that ATL1 molecules carrying S398Y have a defect in the generation of Form 2-like dimer upon GTP binding and are not able to efficiently mediate ER membrane fusion in vivo, implying the physiological importance of loose crossover dimer formation (Fig. 6, Step I). Based on a previous report that most attempts to induce fusion through trans-dimerization of ATL molecules are futile[45,54] and that a successful membrane fusion event requires the cooperation of ATL oligomers formed through their TMDs in each membrane[48], we propose that GTP binding generates several ATL1 molecules with similar loose crossover dimer conformations, allowing them to undergo conformational changes synchronously triggered by GTP hydrolysis for fusion (Fig. 6, Step II and II'). However, if the tight crossover dimers of ATL1 are generated directly from ATL1 molecules in the Apo state with a variety of conformations due to the flexibility in the GTPase domain-3HB region, the driving force for efficient membrane fusion may not be enough.

Another question raised by the finding of GTP-dependent conformational changes in ATL was how the energy released from tight crossover dimer formation is transferred to the membrane. The role of the linker region between the 3HB and TMD in this process was inspired by the crystal structures of ATL1$_{cyto}$ carrying disease mutation N440T, in which the helix of the linker region is disrupted[32]. Our single-molecule FRET imaging assays, combined with MD simulations, further showed that this helical linker region is buried in the lipid bilayer and self-associates in the ATL dimer but is not involved in the nucleotide-dependent assembly of 3HBs. We speculate that the helix formed in the linker region is structurally linked to the energy supply for membrane fusion (Fig. 6, Step II'). A similar mechanism of SNARE-mediated heterotypic membrane fusion has been reported, in which the helix of the linker domain between the SNARE motif and TMD not only provides additional energy[56–58], but also interacts with membranes to facilitate membrane fusion[59]. Interestingly, ATL-mediated homotypic membrane fusion also requires a membrane-destabilizing motif, which is the C-terminal amphipathic helix of ATL[48,51,60].

In summary, our results suggest a comprehensive model for ATL-mediated membrane fusion (Fig. 6) and contribute to understanding the pathogenesis of HSP. The overall mechanisms may be shared by other dynamin-like proteins that act as homotypic fusogens, and the smFRET experiments and MD simulations we performed here can be adapted to further explore their differences.

## Methods
### Reagents
Reagents were obtained from the following sources: Ni Sepharose 6 Fast Flow (Cytiva, 17531802), Glutathione Sepharose 4 Fast Flow (Cytiva, 17513201), tris(2-carboxyethyl)phosphine (Thermo, 75259), EDTA-free protease inhibitor cocktail (Roche, 11873580001), PreScission Protease (Absin, abs01243), Anapoe-X-100 (Anatrace, 9002-93-1), ATP (Roche, 11140965001), MgOAc (Sigma-Aldrich, M0631), L-glutamic acid potassium (Sigma-Aldrich, G1501), biotinylated anti-His antibody (Bioss Antibodies, bs-0287R-bio), streptavidin (Sangon Biotech, A610492), LD555-MAL (Lumidyne, 04), LD655-MAL (Lumidyne, 10), GDP (Sigma-Aldrich, G7127), GTPγS (Roche, 10220647001), GMppNHp (Sigma-Aldrich, G0635), mPEG-SVA-5000 (Laysan Bio, 170-

106), Biotin-PEG-SVA-5000 (Laysan Bio, 170-124), benzoic acid (Sigma-Aldrich, 242381), protocatechuate 3,4-dioxygenase (Sigma-Aldrich, P8279), and n-dodecyl-β-D-maltoside (Avanti Polar Lipids, 850520). All lipids were obtained from Avanti Polar Lipids: 1-palmitoyl-2-oleoyl-glycero-3-phosphocholine (POPC, 850457); 1-palmitoyl-2-oleoyl-sn-glycero-3-phospho-L-serine (POPS, 840034); 1,2-dioleoyl-sn-glycero-3-phosphoethanolamine-N-(7-nitro-2-1,3-benzoxadiazol-4-yl) (NBD-PE, 810144); and 1,2-dioleoyl-sn-glycero-3-phosphoethanolamine-N-(lissamine rhodamine B sulfonyl) (Rhodamine-PE, 810150).

### Strains and plasmids
Yeast strains used in this study were BY4741 (*MATa his3Δ1 leu2Δ2-met15Δ ura3Δ*) and JHY4 (BY4741 *sey1Δ::kanMX4 yop1Δ::HIS3MX6*)[6]. The region encoding the N-terminal cytosolic region of human ATL1 (a.a. 1–447) was cloned into the pET-28a vector using the NcoI and NotI sites. The region encoding the full-length human ATL1 (a.a. 1-558) was cloned into the pRS316 vector along with the endogenous promoter and terminator of *SEY1* using the NotI and XhoI sites. The region encoding the full-length dmATL (a.a. 1-541) was cloned into the pGEX-6p-1 vector using the BamHI and XhoI sites. Point mutations, HA tags and Avi tags were introduced by site-directed mutagenesis.

### Protein expression and purification
ATL1$_{cyto}$ and its mutants were expressed in Rosetta (DE3) (Biomed) *E. coli* cells and purified as described previously[31]. Briefly, cells were harvested and lysed by sonication in purification buffer (25 mM HEPES, pH 7.4, 150 mM NaCl, 5 mM MgCl$_2$, 1X complete EDTA-free protease inhibitor cocktail, 0.5 mM TCEP). The lysates were clarified by centrifugation at 200,000 × g for 1 h at 4 °C using a TYPE 45 Ti rotor. The protein was isolated using an Ni-NTA column and further purified by size exclusion chromatography using Superdex 200 Increase 10/300 GL in purification buffer without protease inhibitor cocktail and TCEP. Fractions containing ATL1$_{cyto}$ were pooled and concentrated.

Full-length dmATL and its mutant were expressed in Rosetta (DE3) (Biomed) *E. coli* cells as described previously[31]. Briefly, cells were harvested by centrifugation and lysed by sonication in purification buffer containing 10% (v/v) glycerol. Membranes were pelleted by centrifugation at 200,000 × g for 1 h at 4 °C using an SW 41 rotor (Beckman Coulter), followed by solubilization in 2.5% (w/v) Anapoe X-100 and 10% (v/v) glycerol in purification buffer for 1 h at 4 °C. The extract was clarified by centrifugation at 200,000 × g for 1 h at 4 °C using an SW 41 rotor and the protein purified from the supernatant by glutathione Sepharose beads. After washing with 0.1% (w/v) Anapoe X-100 and 10% (v/v) glycerol in purification buffer, the GST tag was cleaved by PreScission Protease on a column overnight at 4 °C. The untagged protein was washed with 0.1% (w/v) Anapoe X-100 and 10% (v/v) glycerol in purification buffer.

### Protein biotinylation
For biotinylation, a mixture of 100 μL 100 mM ATP, 100 μL 100 mM MgOAc, 100 μL 500 μM d-biotin, 10 μL 100 μM purified BirA biotin ligase, and 700 μL 50 μM protein was incubated in biotinylation buffer (25 mM HEPES, pH 7.4, 0.5 mM TCEP, 200 mM L-glutamic acid potassium) overnight at 4 °C. Biotinylated protein was incubated with 15 μM streptavidin for 20 min on ice, followed by SDS-PAGE analysis to measure the efficiency of biotinylation. Excess free d-biotin was removed by 10 K MWCO centrifugal filters (Amicon).

### Fluorescence microscopy
Yeast cells were grown to an optical density at 600 nm (OD600) of 1.2-1.5, pelleted, and resuspended in 10% glycerol solution. Resuspended cells were imaged using a super-resolution microscope (DeltaVision OMX SR, Cytiva) with a 60X oil-immersion objective. Sec63p-EGFP was excited by a 488 nm laser. All images were captured by a scientific Complementary Metal Oxide Semiconductor (sCMOS) camera.

## Fluorophore labeling

For intramolecular smFRET assays, ATL1$_{cyto-TK}$ was incubated with LD555 and LD655 at a ratio of 1:1.2:1.2 in labeling buffer (25 mM HEPES, pH 7.4, 150 mM KCl, 5 mM MgCl$_2$) for 5 h at 4 °C. For intermolecular smFRET experiments, ATL1$_{cyto-K}$ was incubated with LD555 or biotinylated ATL1$_{cyto-K}$ with LD655, each at a ratio of 1:1.2 in labeling buffer for 5 h at 4 °C. Excess free fluorophores were removed by Zeba 7 K MWCO spin desalting columns (Thermo Scientific). TCEP was added to the dye-labeled proteins at a final concentration of 0.5 mM.

## Single-molecule FRET imaging and data acquisition

The slide surface of the chamber used in smFRET experiments was modified by the mixture of mPEG-SVA-5000 and Biotin-PEG-SVA-5000 at a ratio of 100:1 in modification buffer (0.1 M NaHCO$_3$ and 0.6 M K$_2$SO$_4$) overnight. Proteins (~2 μM) were incubated with 1 mM GDP, GTPγS, or GDP/AlF$_4^-$ on ice for 1 h, followed by dilution in labeling buffer containing 10 μg/mL streptavidin and 1 mM nucleotide in the chamber to immobilize the dimers (~0.3 nM). For the strategy shown in Fig. 5a, 10 μg/mL streptavidin, 10 μg/mL biotinylated anti-His antibody and 1 mM nucleotide were added to the chamber with dye-labeled ATL1$_{cyto-TK}$ and incubated for 5 min to immobilize the monomers (~1 nM). Buffer containing 2.5 mM benzoic acid, 2 mM protocatechuate 3,4-dioxygenase (PCD) and 1 mM nucleotide was infused into the chamber for oxygen-scavenging before measuring fluorescence measurements.

The imaging was performed using a TIRF microscope (Nikon, Ti2) with a single-frequency 532 nm laser as the excitation source for smFRET experiments and 640 nm laser as the excitation source for photo-bleaching experiments. The microscope was equipped with a high numerical aperture oil immersion objective (Nikon, 100X, N.A. 1.49, oil immersion) to provide high-resolution imaging. To distinguish between the fluorescence signals of the donor and acceptor channels, a 630 nm dichroic mirror was employed. The fluorescence signals were recorded at a frame interval of 30 ms for smFRET experiments and 60 ms for photo-bleaching experiments using an EMCCD camera (Andor, IX897). The raw data were saved in 16-bit TIFF format and further processed for analysis.

## smFRET data analysis

The FRET trajectories were extracted from the recorded movies, and the FRET efficiency was calculated using the equation FRET = I$_{LD655}$/(γI$_{LD555}$ + I$_{LD655}$), where I$_{LD555}$ and I$_{LD655}$ represent the LD555 and LD655 fluorescence intensities, respectively, and γ is the correlation coefficient correcting the difference between the quantum yields and the detection efficiencies of LD555 and LD655. In detail, the parameter γ is equal to the ratio of the reduction in acceptor intensity (F$_A$) to the increase in donor intensity (F$_D$) when acceptor was photobleached (γ = F$_A$/F$_D$). We counted the proportions of hundreds of molecules as a normal distribution in each experiment and regarded the central value of the normal distribution as the final γ value (Supplementary Table 7). The FRET efficiency data from all trajectories were compiled into histograms. These histograms were then fitted into the sum of GaussAmp distributions. The occupancy of each state was estimated by the area under each GaussAmp curve and further used to evaluate the free energy difference between state i and state j using the equation ΔG = -k$_B$Tln(A$_i$/A$_j$)[61,62], where A$_i$ and A$_j$ are the occupancies of state i and state j, respectively, k$_B$ represents the Boltzmann constant, and T is the thermodynamic temperature. We employed a MATLAB script based on the Hidden Markov Model algorithm[63,64] to fit the FRET efficiency trajectories to analyze the dynamics of intramolecular conformational states. The TDPs were derived from the conformational transitions of four-state Hidden Markov Model fitting. The experimental dye-dye distance (R) based on smFRET efficiency (E) was calculated by the following equation: E = 1/[1 + (R/R$_0$)$^6$], in which the estimated Förster radius R$_0$ was 62 Å[47].

## Lipid-mixing assays

The lipid-mixing assays were performed as described previously[31]. Briefly, lipid mixtures of donor liposomes (82:15:1.5:1.5 mole percent of POPC:POPS:NBD-PE:Rhodamine-PE) and acceptor liposomes (85:15 mole percent of POPC:POPS) were dried with N$_2$ to form a film. Purification buffer containing 10% (v/v) glycerol was added to the tube to rehydrate the lipid film, and the suspension underwent 10 freeze-thaw cycles in liquid nitrogen and 37 °C water bath. The liposomes were extruded 21 times through polycarbonate filters with a pore size of 100 nm (Avanti Polar Lipids). The mixture of purified dmATL with liposomes at a protein to lipid ratio of 1:1000 was incubated at 4 °C for 1 h with gentle shaking. The detergent was removed by bio-beads SM-2 resin (Bio-Rad) to reconstitute the proteins into liposomes.

The in vitro lipid-mixing assays were performed in 100 μL volumes. The final lipid concentration was 0.5 mM with donor and acceptor proteoliposomes containing dmATL added at a 1:1 ratio. Reactions were initiated by the addition of 1 mM GTP to the mixtures in a 96-well plate (Corning). The fluorescence intensity of NBD was monitored at 37 °C using an excitation wavelength of 460 nm and emission wavelength of 538 nm every 1 min over 30 min in a Cytation 5 Imaging Reader (BioTek). All data were corrected by subtracting the baseline values obtained at 0 min. The data were expressed as a percentage of the total NBD fluorescence determined by adding 10 μL of 2.5% n-dodecyl-β-D-maltoside to the reactions after 30 min.

## GTPase activity assays

The GTPase assays were performed using the Enzchek Phosphate Assay Kit (Invitrogen) in 100 μL volumes with 5 μL of 20X reaction buffer (1 M Tris, pH 7.5, 20 mM MgCl$_2$, 2 mM sodium azide), 200 mM 2-amino-6-mercapto-7-methylpurine riboside, 1 unit/mL purine nucleoside phosphorylase (PNP), and ATL1$_{cyto}$ or its mutants. Reactions were initiated by the addition of 1 mM GTP in a 96-well plate (Corning). The absorbance at 360 nm was measured at 37 °C every 1 min over 30 min in a Cytation 5 Imaging Reader (BioTek). Data were expressed as absolute absorbance values subtracting the the absorbance at 0 min.

## Molecular dynamics simulations

We modeled the full-length GDP/Pi-bound ATL1 dimer in Form 3 using AlphaFold2 prediction with the AF2complex algorithm[49]. Based on the AF2 prediction and crystal structure of ATL1 bound to GDP (PDB ID: 3Q5E), we further modeled the full-length ATL1 monomer. Next, we inserted these full-length models into a lipid bilayer composed of 80% POPC and 20% POPE, and surrounded them with a water box containing 0.15 M NaCl. The box sizes were 13.5 nm × 13.5 nm × 13.5 nm for ATL1 monomer and 12.1 nm × 12.1 nm × 16.9 nm for Form 3 ATL1 dimer, resulting in ~232,000 and ~250,000 total atoms, respectively. We used the OPM webserver[50] to align the TM region with the lipid bilayer and the CHARMM-GUI webserver[65] to build the systems. We then minimized the energy using the steepest descent algorithm and equilibrated the systems in six steps with gradually reduced position constraints. We ran production simulations in NPT conditions using the CHARMM36m force field[66] for the protein, lipid, GDP, and Pi molecules, and TIP3P for the water molecules. We maintained the temperature at 310 K with a Nosé–Hoover thermostat (1 ps coupling constant) and the pressure at 1.0 bar with a Parrinello–Rahman barostat (5 ps time coupling constant). We used a cut-off of 1.2 nm for van der Waals and short-range electrostatic interactions, with a switch function at 1.0 nm. We used the particle mesh Ewald method with a 0.12 nm mesh spacing for long-range electrostatic interactions. The same methodology was employed to construct models for the N440T mutant of the GDP/Pi-bound Form 3 ATL1 dimer and GDP-bound Form 2 ATL1 dimer. Notably, the N440T mutant model was based on the crystal structure (PDB 4IDQ). We performed two independent simulations, each lasting 1000 ns, for all models using Gromacs 2021.5[67] on GPUs. Trajectories were analyzed by PLUMED[68], mdanalysis[69], and

mdtraj[70]. Theoretical FRET fluorophore distances were calculated from the MD trajectories using FRETpredict[52] and subsequently fitted with a normal distribution using scipy[53].

## Data availability

All underlying data are available from the corresponding authors upon request. The source data underlying Figs. 1d, 2d, 2e, 3c, d, h, i, k, 5c, e and Supplementary Figs. 4, 6f, 8d, 10b, 15 are provided as a Source Data file. The crystal structures of human ATL1$_{cyto}$ and its mutants with different nucleotides have previously been deposited in PDB: 3QOF, 3QNU, 4IDN, 4IDP, 3Q5E, 4IDQ. The molecular dynamics simulations data are available at https://github.com/yongwangCPH/papers/tree/main/2024/ATL1. Source data are provided with this paper.

## Code availability

The custom MATLAB code is available at https://github.com/yangchenguang-1994/HMM-FRET.

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

## Acknowledgements

We thank Yongli Zhang for critical reading of the manuscript. X.B. is supported by the National Natural Science Foundation of China (32371287 and 92154001), the Fundamental Research Funds for the Central Universities (63223043 and 63233053), and the Talent Training Project at Nankai University (035-BB042112). L.M. is supported by the National Natural Science Foundation of China (32271274 and 31770812). Yong W. is supported by the National Key R&D Program of China (2021YFF1200404), the National Natural Science Foundation of China (32371300), and the Fundamental Research Funds for Central Universities (K20220228) and acknowledges access to computational resources from the Information Technology Center and State Key Lab of Computer-Aided Design (CAD) & Computer Graphics (CG) of Zhejiang University. M.L. is supported by National Key R&D Program of China (2019YFA0709304) and National Natural Science Foundation of China (12090051 and T2221001).

## Author contributions

All authors participated in designing the experiments. L.S., K.L., Yunyun W. and X.B. purified the proteins. L.S., C.Y. and K.L. performed smFRET assays. K.L., Yunyun W. and X.B. performed biochemical characterizations of the proteins. L.S., J.L. and R.L. performed functional studies of the proteins in yeast cells. Yong W., M.Z. and K.W. performed MD simulations. L.S., C.Y., M.L., L.M., S.H., M.Z., Yong W. and X.B. analyzed and interpreted data. L.S., C.Y., L.M., S.H. and X.B. initiated the project. L.S., C.Y., L.M., S.H., Yong W. and X.B. wrote the manuscript.

## Competing interests

The authors declare no competing interests.
