## [Peer Review File · Nature Communications]

Dissecting the mechanism of atlastin-mediated homotypic membrane fusion at the single-molecule levelReviewer #1 (Remarks to the Author):

In this study, Shi et al. employ a combination of smFRET and distance-dependent cross-linking experiments to investigate the conformations of the cytoplasmic domain of human ATL1 and its pathogenic mutations across different nucleotide-bound states. The authors' findings not only support the prevailing mechanistic model for ATL functions but also propose an intriguing new working model that presents some deviations from previous hypotheses. While the results are promising, certain aspects warrant further attention and clarification, as outlined below.

Major concerns:

1. The current smFRET setup primarily captures nucleotide-dependent signals, and the interpretation of smFRET data presents challenges. To enhance the credibility of the authors' interpretations, an alternative approach could involve immobilizing one ATL1 and subsequently introducing the other ATL1 in the absence or presence of various nucleotides (GDP, GTP, GMP-PNP). Could this modified smFRET setup offer complementary insights to strengthen the smFRET analysis?
2. In order to reinforce the conclusions drawn from Figure 1, consider performing a control experiment utilizing ATL1-T51C and ATL1-K400C, which could potentially reveal a higher FRET state corresponding to Form 2 or 3 when analyzing intermolecular smFRET. This proposed experiment could also provide corroborating evidence for the interpretation presented in Figure 2.
3. Given that GTPγS is a GTP analog with slow hydrolysis kinetics, substituting it with GMP-PNP, a non-hydrolyzable GTP analog, might yield additional insights. Could the authors comment on their expectations if GMP-PNP were employed instead of GTPγS, particularly in the context of the presented results?
4. The discussion of the GTP-bound ATL1_{cyto} crossover dimers (lines 195-197) raises questions about the existence of both Form 1 and Form 3, considering the FRET signals depicted. A more nuanced explanation of this observation and its implications could enhance the clarity of the discussion.
5. The use of GTPγS as a non-hydrolysable GTP analog in Figures 1 and 2 is noted; however, Figure 3 employs GMP-PNP. Justification for this switch and any potential impact on the overall findings should be addressed.
6. The higher abundance of BMOE-mediated cross-linked dimers in the presence of GDP compared to GMP-PNP demands elucidation. The authors are encouraged to provide insights into the underlying mechanisms or considerations that may contribute to this distinction.
7. Statistical analysis is required to quantify and validate the moderate differences highlighted in Figure 4D. Additionally, a comprehensive discussion elaborating on the physiological significance of loose crossover dimer formation upon GTP binding would enrich the interpretation of these results.
8. In light of recent studies by Crosby et al. (2022) and Jang et al. (2023) demonstrating fusion support by human ATL1 reconstituted into liposomes, the impact of the proline substitution at position 443 on fusion could be assessed using human ATL1, thus enhancing the clinical relevance of the investigation.

The manuscript demonstrates significant potential and offers intriguing insights into the conformational dynamics of ATL1 and its implications in pathogenic mutations. Addressing the above concerns will undoubtedly strengthen the study's impact and contribute to the advancement of the field.

Reviewer #2 (Remarks to the Author):

In this manuscript, Shi et al characterized solution conformational space and ligands-induced changes of the cytosolic region of ATL (ATLcyto), an important membrane protein play crucial role in homotypic membrane fusion, mainly based on smFRET analysis. They claimed that GTP binding induces a loose crossover dimer of ATLcyto (form 2 dimer, a crystallographic dimer), a minor-populated FRET species, and this dimer can be tightened by GTP hydrolysis (mimicked by adding GDP/AlF₄). Based on these observations, authors further found that disease mutation S398Y lowers the population of form 2 dimer in the presence of GTP, and therefore proposed its implication in ATL physiological function. Authors found that disrupting an already known α -helical motif between the ATLcyto-3HB and transmembrane domain (N440T or H443P mutation) has no effect on ATLcyto conformation distribution, but H443P impairs full-length ATL-mediated membrane fusion. MD simulations indicated a role of α -helical motif in protein-membrane interaction. Intramolecular smFRET analysis of ATLcyto monomer revealed the differences of kinetics and dynamics in its apo and GDP bound form, suggesting roles of Pi and GDP release in the regulation of protein recycling process. This topic is interesting, results are technically sound and reported clearly. However, a number of issues are needed to be addressed.

Majors:

1. Authors mostly likely hypothesized that GDP/AlF₄ addition mimics GTP hydrolysis, GDP addition mimics Pi release, and apo state represents GDP release. Do these conditions truly mimic the different states of ATLcyto? It is crucial to clearly state these relationships prior to experimental design and results analysis, and provide evidence or references to support these relationships and ensure their validity.
2. Authors claimed that ATLcyto adopts loose crossover dimer upon (or induced by) GTP binding. "upon" and "induce" are misleading and not correct, authors did not show any smFRET evidence for that before adding GTP, ATLcyto is monomeric or if adopts other dimer conformations. Similar issue for "induced by" used in the first subtitle (line 134), and throughout this manuscript.
3. Fig. 2d, as authors claimed, GTP hydrolysis (GDP/AlF₄ addition) tightens ATLcyto cross dimer, reflected by disappeared Low-FRET species (which is assigned as loose cross dimer by authors). How to understand the existed high-FRET species (major species, which is assigned as tight cross dimer by authors) in the presence of GTP γ S, given that they did not observe states transition under this condition.
3. line 149, issue of "during GTP hydrolysis", addition of GDP/AlF₄ cannot represent the GTP hydrolysis process, it is a steady condition, smFRET were not collected during GTP hydrolysis in real time. Similar description issue throughout this manuscript.
4. Based on known crystal structures, the intramolecular dye labeling Ca-Ca distance of the proposed models in Fig. 1a is 31.8 Å (form 1 dimer), 64.3 Å (form 2 dimer) and 58.2 Å (form 3 dimer), respectively. In parallel, the intermolecular dye labeling Ca-Ca distance in Fig. 2a is 105.5 Å (form 1 dimer), 43.4 Å (form 2 dimer) and 34.4 Å (form 3 dimer), respectively. I would suggest authors to calculate the theoretical dye-dye distances of these proposed models and compare them with the experimentally measured by smFRET. Therefore, each FRET species in Figures 1d, 2d, 4c, 5c, 5f, and particular FRET species in Fig. 7c can be unambiguously assigned. Authors claimed that Fig. 1d intramolecular FRET state cannot be surely assigned as form 2 dimer or form 3 dimer, due to the limited intramolecular smFRET resolution. I do not agree with this, given their ~6 Å dye-dye distance difference (64.3 Å in form 2 dimer and 58.2 Å in form 3 dimer) and the R₀ of the used dyes, the FRET efficiency would differentiate the two conformers. Moreover, similar intermolecular dye-dye distance difference underlies Fig. 2d (9 Å, 43.4 Å in form 2 dimer and 34.4 Å in form 3 dimer), except in this case they are distinguished and assigned as form 2 and form 3 by

authors. More importantly, why two states of intermolecular FRET were observed in the presence of GTP γ S (Fig. 2d), but not monitored in the intramolecular FRET (Fig. 1d), even under the same experimental condition? If Fig. 2d is true, two states should be also observed in Fig 1d. Did authors try other labeling sites? Got similar results? Fig. 2d did not detect form 1 dimer, does it because the dye-dye distance is too far (105.5 Å)? How about using an alternative labeling site?

5. Fig. 3b supports authors' claim that the form 3 tight crossover dimer presents in the presence of either GMppNHp or GDP/AIF4, as a dimer was disulfide linked (5.7 Å distance). However, for Fig. 3c, dimer was only cross-linked by BMOE in presence of GMppNHp but not GDP/AIF4. According to authors' smFRET conclusion, form 2 loose crossover dimer only exists in the presence of non-hydrolyzable GTP analog. Fig. 3c results revealed that BMOE likely specifically cross-linked form 2 loose dimer (11.1 Å distance), but cannot cross link form 3 tight dimer (5.7 Å distance). It does not make sense, it is puzzling that why the BMOE with a maximal arm length of 8 Å can cross link far distance residue (11.1 Å) but cannot cross link short distance residue (5.7 Å)?

6. Authors proposed that the formation of form 2 dimer (by GTP addition) is correlated to ATL physiological function, reflected by slightly decreased population observed for disease mutation S398Y. Does S398Y has impact on membrane fusion (eg. performing Fig. 5d assay for this mutant)? How and / or why S398Y lowers the population of form 2 dimer in the presence of GTP?

7. For another disease mutation, N440T, authors found that it has no impact on ATL conformation distribution, but it may impact ATL function by disrupting a helical motif between 3HB and TMD (formed between N440 and H443). MD simulation indicated helical motifs interaction between the modeled dimer. Generally, to convince the role of helical motif in energy transfer (authors proposed mechanism), I would suggest them to perform additional membrane fusion assay for N440T, additional MD simulations for N440T and K418P mutants, and additionally using form 2 dimer as initial model for MD simulation. More importantly, I highly recommend that the authors utilize full-length ATL for performing smFRET analysis in the membrane environment. This approach would offer more accurate and dynamic insights into ATL and the consequences of mutations, and could also assess the potential movements between soluble domain and transmembrane domain.

Minors:

1. The questions raised by authors in the Introduction (line 99-102), are not actually addressed in this work.

2. Many experiment details are missed. eg. What is the final concentration of GTP γ S, GDP/AIF4 and GDP used in each smFRET experiment? What is the concentration of each component used in the cross-linking assays? What is value of γ for FRET calculation and how it is determined.

3. At a time of 30 ms resolution, intermolecular TIRF-based smFRET captured no states transition. Is there possible fast dynamics beyond this resolution limitation? It might warrant comment in discussion.

4. I would suggest to merge Figures 1 and 2 as a single figure, characterizing conformation spaces of ATLcyto in the presence of GTP γ S, GDP/AIF4; delete Figure 3 or move it to Supplementary figure, because it partially supports authors' claim, or not; merge Figures 4 and 5 as a single figure, characterizing mutation impacts; Figure 6 could also be a Supplementary figure.

6. line 233, I would soften the tone of "dramatically".

7. line 299-301, I would suggest an evidence at single molecular level for that, no dimer is formed. Authors should ensure the traces used for Fig. 7c analysis are single molecular (with one bleaching step). Traces in Fig. 7b did not show the bleaching step, raising the concern that whether they are single molecules.

8. Including the transition kinetics (Supplementary figure 11) and providing a dynamic transition diagram for the monomer ATLcyto in main Figure 7 would be better, and could highlight the difference under the two conditions.

9. Are there references supporting that Pi and GDP are released in sequential manner? Please include them, or prove it.

NCOMMS-23-34444

Rebuttal to the comments raised by the reviewers

We thank the editor and the reviewers for the many constructive comments and suggestions. We have carried out new experiments to address the concerns and made the required additions/modifications to the text.

Key new results include: 1) New smFRET experiments with LD555-ATL1_{cyto-T} and biotinylated LD655-ATL1_{cyto-K} confirming our conclusion regarding the crossover dimer conformation of GTP γ S-bound ATL1_{cyto} molecules, rather than the extended dimer conformation. 2) Molecular Dynamics simulations of the disease mutant disrupting the helical motif between the 3HB and transmembrane domain, confirming that the self-association of these helical motifs in the crossover dimer is essential for the function of ATL1. 3) Demonstration that the disease mutants investigated in this study affect ATL1-mediated ER membrane fusion *in vivo*. 4) Demonstration that the ATL1_{cyto} molecules are monomeric in the nucleotide-free and GDP-bound states and form dimers upon GTP binding and in the transition state of GTP hydrolysis at the single-molecule level in photobleaching experiments. In addition, we have calculated the experimental dye-dye distances based on the smFRET efficiency and estimated the theoretical dye-dye distances using the FRETpredict tool to confirm our proposed assignments. Finally, we have added all of the required details to better describe the assays.

We have combined original Figures 1 and 2 into a single figure, which is now Fig. 1. We have also combined original Figures 4 and 5 into a new figure, which is now Fig. 2. The original Figure 3 was removed as suggested.

The following figures are new: Fig. 2d; Fig. 3f, g and h; Fig. 4c, e and f; Supplementary Fig. 2; Supplementary Fig. 3; Supplementary Fig. 6; Supplementary Fig. 8; Supplementary Fig. 10; Supplementary Fig. 14; Supplementary Fig. 15; Supplementary Table 1-7.

Point to point rebuttal to each point raised by the reviewers.

- Reviewers' text is in black *italics*
- Our responses are in blue

Reviewer #1 (Remarks to the Author):

In this study, Shi et al. employ a combination of smFRET and distance-dependent cross-linking experiments to investigate the conformations of the cytoplasmic domain of human ATL1 and its pathogenic mutations across different nucleotide-bound states. The authors' findings not only support the prevailing mechanistic model for ATL functions but also propose an intriguing new working model that presents some deviations from previous hypotheses. While the results are promising, certain aspects warrant further attention and clarification, as outlined below.

Major concerns:

1. The current smFRET setup primarily captures nucleotide-dependent signals, and the interpretation of smFRET data presents challenges. To enhance the credibility of the authors' interpretations, an alternative approach could involve immobilizing one ATL1 and subsequently introducing the other ATL1 in the absence or presence of various nucleotides (GDP, GTP, GMP-PNP). Could this modified smFRET setup offer complementary insights to strengthen the smFRET analysis?

We thank the reviewer for this great suggestion. As suggested, ~0.5 nM biotinylated ATL1_{cyto-K} molecules labeled with acceptor LD655 were immobilized in the chamber, and ~0.5 nM ATL1_{cyto-K} molecules labeled with donor LD555 were added with or without 1 mM nucleotide. This low protein concentration is required to distinguish the individual fluorescently labeled particles. However, almost no smFRET trajectory of ATL1_{cyto-K} dimer was observed in this experimental system.

We also performed alternating laser excitation (ALEX) experiments, which display the FRET efficiency and the relative stoichiometry of donor and acceptor fluorophores for each single-molecule event, to further monitor the potential nucleotide-dependent ATL1_{cyto-K} dimers under these conditions, but only the acceptor signals were captured in two-dimensional histograms of the FRET and stoichiometry (please see picture below), indicating that no ATL1_{cyto-K} dimers were formed. ALEX is a powerful approach for detecting low FRET events at the edge or low FRET efficiency with a small signal-to-noise ratio. The stoichiometry (S) is calculated by the equation $S = (F_{D|D} + F_{A|D}) / (F_{D|D} + F_{A|D} + F_{A|A})$, where F is the signal intensity, D is donor, A is acceptor, D|D is donor fluorescence upon donor excitation, and similarly for A|D and A|A. Thus, we propose that, at the low protein concentration used in smFRET experiments, the interaction of nucleotide-bound ATL1_{cyto} molecules is not strong enough to form dimer. These results are consistent with a previous study in which almost all of the ATL molecules (0.2 nM) on the

supported bilayer were monomeric in the presence of GTP, GTP γ S, GMPPNP, GDP, or GDP/AIF $_4^-$ (Liu et al, PMID: 25825753).

In the experimental systems used in this study, we mixed the biotinylated LD655-ATL1 $_{\text{cyto-K}}$ and LD555-ATL1 $_{\text{cyto-K}}$ together with or without GDP, GTP γ S, or GDP/AIF $_4^-$ at a high protein concentration of $\sim 2 \mu\text{M}$ to form dimer, and then diluted the protein to $\sim 0.3 \text{ nM}$. The samples were immobilized in a streptavidin-coated microfluidic chamber at this low protein concentration and, according to our results, the nucleotide-dependent dimers were not disassembled during the dilution and immobilization processes. We have now emphasized this dilution step in the revised manuscript.

Two-dimensional histograms of the biotinylated LD655-ATL1 $_{\text{cyto-K}}$ and LD555-ATL1 $_{\text{cyto-K}}$ molecules incubated at a low protein concentration of 0.5 nM in the absence or presence of GTP γ S, GDP/AIF $_4^-$, or GDP.

2. In order to reinforce the conclusions drawn from Figure 1, consider performing a control experiment utilizing ATL1-T51C and ATL1-K400C, which could potentially reveal a higher FRET state corresponding to Form 2 or 3 when analyzing intermolecular smFRET. This proposed experiment could also provide corroborating evidence for the interpretation presented in Figure 2.

We thank the reviewer for this excellent point to provide additional evidence for the model we proposed. We have performed the control experiment as suggested. Specifically, ATL1 $_{\text{cyto-T}}$ labeled with donor LD555 and biotinylated ATL1 $_{\text{cyto-K}}$ labeled with acceptor LD655 were incubated with GTP γ S or GDP/AIF $_4^-$ to form ATL1 $_{\text{cyto-T}}$ -ATL1 $_{\text{cyto-K}}$ dimer (new Supplementary Fig. 8a). The dimer was subsequently diluted and immobilized in a streptavidin-coated microfluidic chamber (new Supplementary Fig. 8b). The ATL1 $_{\text{cyto-T}}$ -ATL1 $_{\text{cyto-K}}$ dimer in the GTP γ S-bound and GDP/AIF $_4^-$ -bound state presented a high stable FRET state centered at ~ 0.80 (new Supplementary Fig. 8c and d), consistent with the short distance between the two labeled sites in the Form 2 and Form 3 crossover dimers (new Supplementary Table 2 and 3). These data confirm our conclusion that GTP binding to ATL1 $_{\text{cyto}}$ does not generate a Form 1 extended dimer, but induces crossover dimer formation.

3. Given that GTP γ S is a GTP analog with slow hydrolysis kinetics, substituting it with GMP-PNP, a non-hydrolyzable GTP analog, might yield additional insights.

Could the authors comment on their expectations if GMP-PNP were employed instead of GTPγS, particularly in the context of the presented results?

In our previous study (Table S3, Bian et al, PMID: 21368113), the binding of GTPγS and GDP to ATL1_{cyto} was measured by isothermal titration calorimetry (ITC), and the equilibrium dissociation constants (K_D) were 1.15 and 1.84 μM, respectively. However, the binding affinity of GMPPNP to ATL1_{cyto} was lower ($K_D=12.89$ μM, please see picture below). Thus, GMPPNP may not be a good GTP analog for studying the structural dynamics of ATL at this low protein concentration in smFRET assays.

Binding affinity of GMPPNP to ATL1_{cyto}. The estimated number of binding sites (N) and the dissociation constants K_d are shown.

4. The discussion of the GTP-bound ATL1_{cyto} crossover dimers (lines 195-197) raises questions about the existence of both Form 1 and Form 3, considering the FRET signals depicted. A more nuanced explanation of this observation and its implications could enhance the clarity of the discussion.

We thank the reviewer for pointing out the necessity to clarify this point. We have revised the statement as follows: “Given that the FRET efficiency negatively correlates with the distance between the donor and acceptor fluorophores and no Form 1 extended dimer was formed in the presence of GTPγS or GDP/AlF₄⁻ (Fig. 1a-

d and Supplementary Fig. 8), these data demonstrate that the GTP-bound ATL1_{cyto} crossover dimers adopt two conformations corresponding to the loose crossover dimer and Form 3 tight crossover dimer, respectively, and the tight association of 3HBs is completely formed in the transition state of GTP hydrolysis”

5. The use of GTPγS as a non-hydrolysable GTP analog in Figures 1 and 2 is noted; however, Figure 3 employs GMP-PNP. Justification for this switch and any potential impact on the overall findings should be addressed.

We performed new crosslinking assays with or without different nucleotides, including GMPPNP, GMPPCP, GTPγS, GDP/AlF₄⁻, and GDP (please see picture below). As expected, in the presence of GMPPNP or GTPγS, both BMOE and diamide induced cross-linked dimer formation. These data support the idea that ATL1_{cyto} adopts either a loose or tight crossover dimer upon GTP binding. Interestingly, no crosslinking was observed with GMPPCP, indicating that it cannot mimic GTP for ATL.

As suggested by Reviewer #2, we have removed the original Figure 3.

Purified ATL1_{cyto}-L357C was incubated with BMOE or oxidant diamide in the absence or presence of the indicated nucleotide. The dimer is indicated by the black arrow.

6. The higher abundance of BMOE-mediated cross-linked dimers in the presence of GDP compared to GMP-PNP demands elucidation. The authors are encouraged to provide insights into the underlying mechanisms or considerations that may contribute to this distinction.

At high protein concentration, the GDP-bound ATL1_{cyto} has been reported to form a weak dimer (Figure 8, Liu et al, PMID: 25825753). As its conformation is very dynamic, which was shown in our manuscript, we propose that the loose crossover

dimer is captured by BMOE in the presence of GDP during the incubation period. In the presence of GTPγS, though its dimerization is much stronger, its conformation is relatively stable and only a proportion of GTPγS-bound ATL1_{cyto} molecules adopt a loose crossover dimer. Consequently, it is very likely that the BMOE-crosslinked ATL1_{cyto} dimer in the presence of GDP is more abundant than with GTPγS.

7. Statistical analysis is required to quantify and validate the moderate differences highlighted in Figure 4D.

We performed a statistical analysis and provide the results in new Fig. 2d.

Additionally, a comprehensive discussion elaborating on the physiological significance of loose crossover dimer formation upon GTP binding would enrich the interpretation of these results.

To emphasize this point, we now state: “Via a combination of smFRET analyses and imaging experiments using yeast cells, our results indicated that ATL1 molecules carrying S398Y have a defect in the generation of Form 2-like dimer upon GTP binding and are not able to efficiently mediate ER membrane fusion in vivo, implying the physiological importance of loose crossover dimer formation (Fig. 5, Step I). Based on a previous report that most attempts to induce fusion through trans-dimerization of ATL molecules are futile^{45, 52} and that a successful membrane fusion event requires the cooperation of ATL oligomers formed through their TMDs in each membrane⁴⁸, we propose that GTP binding generates several ATL1 molecules with similar loose crossover dimer conformations, allowing them to undergo conformational changes synchronously triggered by GTP hydrolysis for fusion (Fig. 5, Step II and II’). However, if the tight crossover dimers of ATL1 are generated directly from ATL1 molecules in the Apo state with a variety of conformations due to the flexibility in the GTPase domain-3HB region, the driving force for efficient membrane fusion may not be enough.”

8. In light of recent studies by Crosby et al. (2022) and Jang et al. (2023) demonstrating fusion support by human ATL1 reconstituted into liposomes, the impact of the proline substitution at position 443 on fusion could be assessed using human ATL1, thus enhancing the clinical relevance of the investigation.

We thank the reviewer for this great suggestion. To directly validate the physiological significance of the mutations investigated in this study, we performed imaging experiments using yeast cells, a well-established system for testing the function of human ATL1, its plant ortholog RHD3, and other dynamin-like membrane fusion protein that is artificially anchored to the ER membrane (Anwar et al, PMID: 22508509; Liu et al, PMID: 22802620; Chen et al, PMID: 21652628; Zhang et al, PMID: 23922269; Huang et al, PMID: 29093165). In this assay, yeast cells lacking Sey1p, a functional ortholog of ATL, and another ER-shaping protein, Yop1p, had a

significant reduction in the intricate tubular ER network at the periphery of the cell due to a defect in ER membrane fusion. The expression of WT ATL1 could rescue this abnormal cortical ER morphology (new Supplementary Fig. 10). However, in the *sey1Δyop1Δ* cells expressing disease mutant N440T or S398Y, or helix-disrupting mutant H443P, the restoration of the tubular ER network to the WT levels was not observed (new Supplementary Fig. 10). These results indicate that these mutations affect the membrane fusion mediated by ATL1.

The manuscript demonstrates significant potential and offers intriguing insights into the conformational dynamics of ATL1 and its implications in pathogenic mutations. Addressing the above concerns will undoubtedly strengthen the study's impact and contribute to the advancement of the field.

We thank the reviewer for the positive comments on our manuscript.

Reviewer #2 (Remarks to the Author):

In this manuscript, Shi et al characterized solution conformational space and ligands-induced changes of the cytosolic region of ATL (ATL_{cyto}), an important membrane protein play crucial role in homotypic membrane fusion, mainly based on smFRET analysis. They claimed that GTP binding induces a loose crossover dimer of ATL_{cyto} (form 2 dimer, a crystallographic dimer), a minor-populated FRET species, and this dimer can be tightened by GTP hydrolysis (mimicked by adding GDP/AIF4). Based on these observations, authors further found that disease mutation S398Y lowers the population of form 2 dimer in the presence of GTP, and therefore proposed its implication in ATL physiological function. Authors found that disrupting an already known α -helical motif between the ATL_{cyto}-3HB and transmembrane domain (N440T or H443P mutation) has no effect on ATL_{cyto} conformation distribution, but H443P impairs full-length ATL-mediated membrane fusion. MD simulations indicated a role of α -helical motif in protein-membrane interaction. Intramolecular smFRET analysis of ATL_{cyto} monomer revealed the differences of kinetics and dynamics in its apo and GDP bound form, suggesting roles of Pi and GDP release in the regulation of protein recycling process. This topic is interesting, results are technically sound and reported clearly. However, a number of issues are needed to be addressed.

Majors:

1. Authors mostly likely hypothesized that GDP/AIF4 addition mimics GTP hydrolysis, GDP addition mimics Pi release, and apo state represents GDP release. Do these conditions truly mimic the different states of ATL_{cyto}? It is crucial to clearly state these relationships prior to experimental design and results analysis, and provide evidence or references to support these relationships and ensure their validity.

According to previous studies (Ahmadian et al, PMID: 9188784; Praefcke et al, PMID: 10493878), AIF_4^- associates with GTPase only in its GDP-bound state, mimicking the transition state of the GTP hydrolysis reaction. For $\text{ATL1}_{\text{cyto}}$, these conditions have been widely used to mimic the different states of ATL to investigate the different temporal steps in the GTPase cycle (Liu et al, PMID: 25825753; Uleengin et al, PMID: 25761634; O'Donnell et al, PMID: 28602821). We have now added these references to the revised manuscript.

2. Authors claimed that ATL_{cyto} adopts loose crossover dimer upon (or induced by) GTP binding. "upon" and "induce" are misleading and not correct, authors did not show any smFRET evidence for that before adding GTP, ATL_{cyto} is monomeric or if adopts other dimer conformations. Similar issue for "induced by" used in the first subtitle (line 134), and throughout this manuscript.

We thank the reviewer for this excellent point to provide evidence that nucleotide-free $\text{ATL1}_{\text{cyto}}$ is monomeric. We performed photobleaching experiments using LD655-labeled $\text{ATL1}_{\text{cyto-K}}$ to test the dimerization state of the protein at the single-molecule level. The C-terminal Avi tag of LD655- $\text{ATL1}_{\text{cyto-K}}$ was biotinylated. The modified proteins were mixed with or without GDP, $\text{GTP}\gamma\text{S}$, or $\text{GDP}/\text{AIF}_4^-$ at a high protein concentration of $\sim 2 \mu\text{M}$, followed by a dilution and immobilization of the protein ($\sim 0.3 \text{ nM}$) in a streptavidin-coated microfluidic chamber (new Supplementary Fig. 6a and b). The samples were imaged by TIRF microscopy. At this low protein concentration, the individual fluorescently labeled particles could be distinguished, and almost all of them presented one-step photobleaching in the absence or presence of GDP (new Supplementary Fig. 6c and e). With $\text{GTP}\gamma\text{S}$ or $\text{GDP}/\text{AIF}_4^-$, the fluorescence of most LD655- $\text{ATL1}_{\text{cyto-K}}$ molecules was photobleached in two steps (Supplementary Fig. 6d and e). These data show that the $\text{ATL1}_{\text{cyto}}$ molecules are monomeric in the nucleotide-free and GDP-bound states and form dimers in the presence of $\text{GTP}\gamma\text{S}$ or $\text{GDP}/\text{AIF}_4^-$ under the conditions of single-molecule experiments, which is consistent with previous bulk FRET experiments (Liu et al, PMID: 25825753; O'Donnell et al, PMID: 28602821).

We also performed alternating laser excitation (ALEX) experiments, which display the FRET efficiency and relative stoichiometry of donor and acceptor fluorophores for each single-molecule event, to further monitor the potential nucleotide-dependent dimerization between biotinylated LD655- $\text{ATL1}_{\text{cyto-K}}$ and LD555- $\text{ATL1}_{\text{cyto-K}}$ in the absence or presence of different nucleotides. Only the acceptor LD655 signals were captured in the Apo state in two-dimensional histograms of the FRET and stoichiometry (please see picture below). ALEX is a powerful approach for detecting low FRET events at the edge or low FRET efficiency with a small signal-to-noise ratio. The stoichiometry (S) is calculated by the equation $S = (F_{\text{D|D}} + F_{\text{A|D}}) / (F_{\text{D|D}} + F_{\text{A|D}} + F_{\text{A|A}})$, where F is the signal intensity, D is donor, A is acceptor, D|D is donor fluorescence upon donor excitation, and similarly for A|D and A|A. These results imply that no $\text{ATL1}_{\text{cyto}}$ dimers were formed without nucleotide.

Two-dimensional histograms of the biotinylated LD655-ATL1_{cyto-K} and LD555-ATL1_{cyto-K} molecules incubated at a high protein concentration of 2 μ M in the absence or presence of GTP γ S, GDP/AIF₄, or GDP.

3. Fig. 2d, as authors claimed, GTP hydrolysis (GDP/AIF₄ addition) tightens ATL_{cyto} cross dimer, reflected by disappeared Low-FRET species (which is assigned as loose cross dimer by authors). How to understand the existed high-FRET species (major species, which is assigned as tight cross dimer by authors) in the presence of GTP γ S, given that they did not observe states transition under this condition.

We understand the point raised by the reviewer. We note that we did observe a very small number of smFRET traces reflecting transitions between low- and high-FRET states for GTP γ S-bound ATL1_{cyto} dimers (please see picture below). We propose that, in the presence of GTP γ S, ATL1_{cyto} molecules undergo low-frequency transition between loose crossover dimer and tight crossover dimer on a sub-millisecond timescale, and this fast transition could not be efficiently captured by our smFRET setup (30 ms per frame). We make this point more explicitly in the revised manuscript.

Fluorescence and smFRET trajectory of intermolecular smFRET assay for ATL1_{cyto-K} dimer in the presence of GTP γ S.

3. line 149, issue of "during GTP hydrolysis", addition of GDP/AIF₄ cannot represent the GTP hydrolysis process, it is a steady condition, smFRET were not collected during GTP hydrolysis in real time. Similar description issue throughout this manuscript.

We thank the reviewer for pointing this out. We have revised the statement as follows: “To assess the conformation of a protomer in the ATL1_{cyto} dimer upon GTP binding or in the transition state of GTP hydrolysis using intramolecular smFRET, we labeled ATL1_{cyto-TK} with the LD555/LD655 fluorophore pair⁴⁷ (Fig. 1a) and biotinylated the C-terminal Avi-tagged ATL1_{cyto-K} (Supplementary Fig. 5).” We have also changed the descriptions throughout the manuscript.

4. Based on known crystal structures, the intramolecular dye labeling Ca-Ca distance of the proposed models in Fig. 1a is 31.8 Å (form 1 dimer), 64.3 Å (form 2 dimer) and 58.2 Å (form 3 dimer), respectively. In parallel, the intermolecular dye labeling Ca-Ca distance in Fig. 2a is 105.5 Å (form 1 dimer), 43.4 Å (form 2 dimer) and 34.4 Å (form 3 dimer), respectively. I would suggest authors to calculate the theoretical dye-dye distances of these proposed models and compare them with the experimentally measured by smFRET. Therefore, each FRET species in Figures 1d, 2d, 4c, 5c, 5f, and particular FRET species in Fig. 7c can be unambiguously assigned.

We thank the reviewer for this suggestion. The experimental dye-dye distance (R) based on the smFRET efficiency (E) was calculated using the equation $E = 1/[1+(R/R_0)^6]$, with the estimated R_0 (62 Å, Girodat et al, PMID: 33151943). We also employed the FRETpredict tool (<https://pypi.org/project/FRETpredict/>; Montepietra et al, doi: <https://doi.org/10.1101/2023.01.27.525885>, bioRxiv; Virtanen et al, PMID: 32015543) to estimate both the intermolecular and intramolecular dye-dye distances, leveraging two repeat runs of the MD simulation trajectories for Form 1, Form 2, and Form 3. These values for the theoretically and experimentally estimated dye-dye distances, as detailed in new Supplementary Table 1-4 and 6, exhibit an offset of approximately 10-30 Å compared to the C α -C α distances calculated from the crystal structures. Despite some discrepancies between the theoretical and experimental estimations, which could be attributed to uncertainties in the Förster radius R_0 and limited MD sampling, our findings consistently support the proposed assignments.

Authors claimed that Fig. 1d intramolecular FRET state cannot be surely assigned as form 2 dimer or form 3 dimer, due to the limited intramolecular smFRET resolution. I do not agree with this, given their ~6 Å dye-dye distance difference (64.3 Å in form 2 dimer and 58.2 Å in form 3 dimer) and the R_0 of the used dyes, the FRET efficiency would differentiate the two conformers. Moreover, similar intermolecular dye-dye distance difference underlies Fig. 2d (9 Å, 43.4 Å in form 2 dimer and 34.4 Å in form 3 dimer), except in this case they are distinguished and assigned as form 2 and form 3 by authors. More importantly, why two states of intermolecular FRET were observed in the presence of GTP γ S (Fig. 2d), but not monitored in the intramolecular FRET (Fig. 1d), even under the same experimental condition? If Fig. 2d is true, two states should be also observed in Fig 1d. Did authors try other labeling sites? Got similar results?

According to the theoretical dye-dye distances calculated by the FRETpredict tool (new Supplementary Table 2), the intramolecular dye-dye distances of ATL1_{cyto-TK} ($R_1 = 86 \pm 5 \text{ \AA}$ in Form 2 and $R_2 = 82 \pm 4 \text{ \AA}$ in Form 3), which are considerably larger than the estimated R_0 (62 \AA , Girodat et al, PMID: 33151943), are expected to be indistinguishable in smFRET (theoretical $\Delta E_{2-1} = \sim 0.034$). More significantly, the intermolecular dye distances of ATL1_{cyto-K} dimer exhibit notable differences between Form 2 and Form 3, with measurements of $R_3 = 65 \pm 2 \text{ \AA}$ for Form 2 and $R_4 = 53 \pm 3 \text{ \AA}$ for Form 3 (theoretical $\Delta E_{4-3} = \sim 0.290$, new Supplementary Table 2). This underscores the pivotal role of the FRET efficiency of intermolecular ATL1_{cyto-K} dimer in distinguishing between these two conformations. We appreciate the reviewer's insightful input, and these analyses contribute to the robustness of our conclusions.

We have also performed new intermolecular smFRET experiments using LD555-ATL1_{cyto-T} and biotinylated LD655-ATL1_{cyto-K}. These two modified molecules were incubated together with GTP γ S or GDP/AIF₄⁻ to form ATL1_{cyto-T}-ATL1_{cyto-K} dimer (new Supplementary Fig. 8a). The dimer was subsequently diluted and immobilized in a streptavidin-coated microfluidic chamber (new Supplementary Fig. 8b). ATL1_{cyto-T}-ATL1_{cyto-K} dimer in the GTP γ S-bound and GDP/AIF₄⁻-bound state had a single high-FRET state centered at ~ 0.80 (new Supplementary Fig. 8c and d), consistent with the short distance between the two labeled sites in the Form 2 ($R_5 = 51 \pm 4 \text{ \AA}$) and Form 3 ($R_6 = 50 \pm 4 \text{ \AA}$) crossover dimers (theoretical $\Delta E_{6-5} = \sim 0.021$, new Supplementary Table 2).

Fig. 2d did not detect form 1 dimer, does it because the dye-dye distance is too far (105.5 Å)? How about using an alternative labeling site?

As shown in the response to Major Point 2, we performed ALEX experiments to demonstrate that the ATL1_{cyto} dimer existed in two forms (Form 2 and Form 3 crossover dimer) in the presence of GTP γ S and that there was no Form 1 extended dimer formation. As described above, we also performed new intermolecular smFRET experiments using LD555-ATL1_{cyto-T} and biotinylated LD655-ATL1_{cyto-K}. The results support our conclusion that GTP binding to ATL1_{cyto} does not generate a Form 1 extended dimer, but induces crossover dimer formation (new Supplementary Fig. 8).

5. Fig. 3b supports authors' claim that the form 3 tight crossover dimer presents in the presence of either GMppNHp or GDP/AIF₄, as a dimer was disulfide linked (5.7 Å distance). However, for Fig. 3c, dimer was only cross-linked by BMOE in presence of GMppNHp but not GDP/AIF₄. According to authors' smFRET conclusion, form 2 loose crossover dimer only exists in the presence of non-hydrolyzable GTP analog. Fig. 3c results revealed that BMOE likely specifically cross-linked form 2 loose dimer (11.1 Å distance), but cannot cross link form 3 tight dimer (5.7 Å distance). It does not make sense, it is puzzling that why the BMOE with a maximal arm length of 8 Å

can cross link far distance residue (11.1 Å) but cannot cross link short distance residue (5.7 Å)?

We agree with the reviewer that the result of no or little BMOE-mediated conjugation between two C357 residues in the tight crossover dimer is puzzling. However, as the behavior of ATL1_{cyto} with GTPγS was different from the behavior with GDP/AlF₄⁻ in the crosslinking assays, these data are in agreement with our conclusion that the conformations of GTPγS-bound and GDP/AlF₄⁻-bound ATL1_{cyto} molecules are distinct.

We removed the original Figure 3 as suggested in Minor Point 4.

6. Authors proposed that the formation of form 2 dimer (by GTP addition) is correlated to ATL physiological function, reflected by slightly decreased population observed for disease mutation S398Y. Does S398Y has impact on membrane fusion (eg. performing Fig. 5d assay for this mutant)?

We thank the reviewer for this valuable suggestion. To directly confirm the physiological significance of the mutations investigated in this study, we performed imaging experiments using yeast cells, a well-established system for testing the function of human ATL1, its plant ortholog RHD3, and other dynamin-like membrane fusion protein that is artificially anchored to the ER membrane (Anwar et al, PMID: 22508509; Liu et al, PMID: 22802620; Chen et al, PMID: 21652628; Zhang et al, PMID: 23922269; Huang et al, PMID: 29093165). In this assay, yeast cells lacking Sey1p (a functional ortholog of ATL) and another ER-shaping protein, Yop1p, had a significant reduction in the intricate tubular ER network at the periphery of the cell due to a defect in ER membrane fusion. The expression of WT ATL1 could rescue this abnormal cortical ER morphology (Supplementary Fig. 10). However, in the *sey1Δyop1Δ* cells expressing disease mutant N440T or S398Y, or helix-disrupting mutant H443P, restoration of the tubular ER network to the WT levels was not observed (Supplementary Fig. 10). These results indicate that these mutations affect the membrane fusion mediated by ATL1.

How and / or why S398Y lowers the population of form 2 dimer in the presence of GTP?

We propose that S398Y has an impact on the hydrophobic interactions within the three bundled α-helices. Thus, S398Y may give rise to a new 3HB topology, which is prone to tightly self-associate. The structure of ATL1-S398Y in the presence of different nucleotides deserves further investigation.

7. For another disease mutation, N440T, authors found that it has no impact on ATL conformation distribution, but it may impact ATL function by disrupting a helical motif between 3HB and TMD (formed between N440 and H443). MD simulation indicated

helical motifs interaction between the modeled dimer. Generally, to convince the role of helical motif in energy transfer (authors proposed mechanism), I would suggest them to perform additional membrane fusion assay for N440T,

Please see our response to Major Point 6. We performed imaging assays using *sey1Δyop1Δ* yeast cells to confirm that the disease mutants investigated in this study affect ATL1-mediated ER membrane fusion (new Supplementary Fig. 10).

additional MD simulations for N440T and K418P mutants, and additionally using form 2 dimer as initial model for MD simulation.

We thank the reviewer for these insightful suggestions. We carried out additional simulations for both the H443P and N440T mutants in the GDP/Pi-bound Form 3 dimer. The H443P mutant model was still based on the crystal structure of WT ATL1_{cyto} (PDB 4IDO). Our simulations revealed a noticeable decrease in hydrogen-bonding interactions within the helical motif, indicating reduced helicity in the region affected by the H443P mutation (please see picture below). However, because the helical motif did not completely lose its structure within the 1- μ s simulation, likely due to the limited simulation time scale, we did not observe a robust reduction in interactions between the helical motifs. For the N440T mutant, simulations initiated from the crystal structure of ATL1_{cyto}-N440T (PDB 4IDQ), wherein the helical motif is almost unstructured, revealed significant decreases in hydrogen-bonding interactions within the helical motif (please see picture below) and interactions between the helical motifs (new Fig. 3g and h). We also followed the reviewer's suggestion to explore the GDP-bound Form 2 dimer as an initial model for MD simulation. The results demonstrated that the helical motifs in the dimer, which were initially separated, gradually approached and interacted with each other during the 1- μ s simulation (new Supplementary Fig. 14). However, the 3HBs did not move closer within this time scale, which confirms that the assembly of the 3HBs and the interaction between the helical motifs are independent and this helical motif has strong potential to self-associate. These additional simulations support our proposed mechanism.

Hydrogen-bonding interactions within the helical motif.

More importantly, I highly recommend that the authors utilize full-length ATL for performing smFRET analysis in the membrane environment. This approach would offer more accurate and dynamic insights into ATL and the consequences of mutations, and could also assess the potential movements between soluble domain and transmembrane domain.

We agree with the reviewer that using full-length ATL for smFRET experiments could give us more information on the conformational dynamics of the protein. We attempted to purify the full-length Cys-less ATL1-K400C (four inherent cysteines in the N-terminal cytosolic region, one in the transmembrane domain, and one in the C-terminal tail). Unfortunately, we did not obtain the purified protein for labeling. We also attempted to introduce an unnatural amino acid for labeling, but the system did not function. Despite our extensive experience in using different detergents to purify various membrane proteins, we encountered challenges in this endeavor. Developing a strategy for fluorescently labeling full-length ATL will be one of our major research directions in the future.

Minors:

1. The questions raised by authors in the Introduction (line 99-102), are not actually addressed in this work.

We shortened this part. We now state: "In addition to the controversy regarding domain rearrangements upon GTP binding and GTP hydrolysis, the ATL monomer conformations regulated by the sequential release of Pi and GDP⁴⁶ to start a new round of fusion are still poorly understood."

2. Many experiment details are missed. eg. What is the final concentration of GTP γ S, GDP/AIF4 and GDP used in each smFRET experiment? What is the concentration of each component used in the cross-linking assays? What is value of γ for FRET calculation and how it is determined.

The final concentration of GTP γ S, GDP/AIF₄⁻ or GDP used in smFRET experiments was 1 mM.

In smFRET experiments, the fluorescence transfer efficiency (E) between donor and acceptor is usually calculated by the formula $E = I_A / (I_D + I_A)$. It is feasible to use this formula when the quantum yields and detection efficiencies of the donor and acceptor are the same. However, donor and acceptor quantum yields and detection efficiencies vary with experimental conditions, so it is necessary to introduce parameter γ to obtain an accurate efficiency E via the formula $E = I_A / (\gamma I_D + I_A)$. The parameter γ is equal to the ratio of the reduction in acceptor intensity (F_A) to the increase in donor intensity (F_D) when the acceptor was photobleached ($\gamma = F_A / F_D$, please see picture below). We counted the proportions of hundreds of molecules as

a normal distribution in each experiment and regarded the central value of the normal distribution as the final γ value.

We added these details to the revised manuscript and provide the γ values for each set of experiments in the new Supplementary Table 7.

Determination of smFRET correlation coefficient γ via acceptor photobleaching. Representation of F_D and F_A in the intermolecular (a) or intramolecular (b) trajectory. Correlation coefficient γ was calculated by the formula $\gamma = F_A / F_D$.

3. At a time of 30 ms resolution, intermolecular TIRF-based smFRET captured no states transition. Is there possible fast dynamics beyond this resolution limitation? It might warrant comment in discussion.

As mentioned in the response to Major Point 3, we cannot rule out the possibility that GTP γ S-bound ATL1_{cyto} dimers undergo a low-frequency transition between low- and high-FRET states on a sub-millisecond timescale, and this fast transition could not be efficiently captured by our smFRET setup (30 ms per frame). We made this point more explicitly in the revised manuscript.

4. I would suggest to merge Figures 1 and 2 as a single figure, characterizing conformation spaces of ATL_{cyto} in the presence of GTP γ S, GDP/AIF4; delete Figure 3 or move it to Supplementary figure, because it partially supports authors' claim, or not; merge Figures 4 and 5 as a single figure, characterizing mutation impacts; Figure 6 could also be a Supplementary figure.

Thank you for pointing this out. We combined Fig. 1 and 2 as new Fig. 1, and Fig 4 and 5 as new Fig. 2. The Fig. 3 has been removed as suggested. However, we still retained Fig. 6 (new Fig. 3) as a main figure, as we believe it provides additional insights into the mechanism of ATL-mediated membrane fusion.

6. line 233, I would soften the tone of "dramatically".

We deleted "dramatically".

7. line 299-301, I would suggest an evidence at single molecular level for that, no dimer is formed. Authors should ensure the traces used for Fig. 7c analysis are

single molecular (with one bleaching step). Traces in Fig. 7b did not show the bleaching step, raising the concern that whether they are single molecules.

As explained in the response to Major Point 2, we performed new photobleaching experiments and ALEX experiments to demonstrate that the nucleotide-free and GDP-bound ATL1_{cyto} molecules are monomeric at the single-molecule level. We have also given the representative trajectories a sufficient length of time in new Fig. 4b to show that the fluorescence was photobleached in a single step. Thus, the data we fit in new Fig. 4c were from single molecules.

8. Including the transition kinetics (Supplementary figure 11) and providing a dynamic transition diagram for the monomer ATL_{cyto} in main Figure 7 would be better, and could highlight the difference under the two conditions.

We modified the original Fig. 7 (new Fig. 4), show the transition rate constants in the original Supplementary Fig. 11 (new Supplementary Fig. 15), and added a figure (new Fig. 4e) summarizing the differences in the transition rate constants between GDP-bound ATL1_{cyto} and nucleotide-free ATL1_{cyto}.

9. Are there references supporting that Pi and GDP are released in sequential manner? Please include them, or prove it.

The reaction scheme of GTP hydrolysis by the GTPase has been studied extensively and is summarized in the following stages (Sprang, PMID: 9242920): (1) formation of the GTPase-GTP complex, (2) generation of the transition state by bond cleavage, (3) release of the γ -phosphate, and (4) exchange of GDP for GTP. We included this reference in the revised manuscript.

Reviewer #1 (Remarks to the Author):

I am satisfied with the authors' responses to the issues that I brought up. Now I'd like to recommend this revised manuscript for publication in Nature Communications.

Reviewer #2 (Remarks to the Author):

This revised manuscript has addressed most of my concerns and provided additional results to reinforce the proposed mechanism. There are some minor suggestions to authors.

1. Supplementary Fig. 6, an image picture could be included to exhibit the single molecule distribution.
2. Regarding to authors' smFRET experiment setup, despite the exactly concentrations of individual ligands in the chamber are not known, for Fig. 1h, whether higher GTPyS concentration (> 1 mM) would shift the low-FRET species to 100% high-FRET?
3. Authors used "ligand-bound" throughout this manuscript (like GDP-bound, GTPyS-bound, GDP/AlF₄-bound, etc.) when interpreting the smFRET data, however it is unsure that the observed single molecules are indeed in the ligand-bound form. The FRET signal only reflects the protein behaviors in the presence of ligand. I would suggest to revise these depictions.
4. Authors' comments on the discrepancies between the theoretical and experimental distances should be included in the manuscript.

NCOMMS-23-34444A

Rebuttal to the comments raised by the reviewers

We have addressed the remaining concerns and made the required changes to the text and figures.

Because the figure legends should not exceed 350 words, we have to split the original Fig. 1 in two (new Fig. 1 and Fig. 2).

The following figures are new: Fig. 2e, Fig. 3d, Supplementary Fig. 6c, Supplementary Fig. 14.

Point to point rebuttal to each point raised by the reviewers.

-Reviewers' text is in black *italics*

-Our responses are in blue

Reviewer #2 (Remarks to the Author):

This revised manuscript has addressed most of my concerns and provided additional results to reinforce the proposed mechanism. There are some minor suggestions to authors.

1. Supplementary Fig. 6, an image picture could be included to exhibit the single molecule distribution.

We have provided these data in the new Supplementary Fig. 6c.

2. Regarding to authors' smFRET experiment setup, despite the exactly concentrations of individual ligands in the chamber are not known, for Fig. 1h, whether higher GTP γ S concentration (> 1 mM) would shift the low-FRET species to 100% high-FRET?

We realize that we did not make sufficiently clear that all the buffer we used in the smFRET imaging contained 1 mM nucleotide. Because the GTP γ S concentration (1mM) was much higher than that of the immobilized protein (< 0.3 nM) in the chamber, we do not expect that a higher concentration of GTP γ S would change the FRET distribution. We have now made this point more explicitly in Methods.

3. Authors used "ligand-bound" throughout this manuscript (like GDP-bound, GTP γ S-bound, GDP/AIF4-bound, etc.) when interpreting the smFRET data, however it is unsure that the observed single molecules are indeed in the ligand-bound form. The FRET signal only reflects the protein behaviors in the presence of ligand. I would suggest to revise these depictions.

We have changed the descriptions to "in the presence of ligand" throughout the manuscript.

4. Authors' comments on the discrepancies between the theoretical and experimental distances should be included in the manuscript.

We have included the comments on the discrepancies between the theoretical and experimental dye-dye distances in Discussion in the revised manuscript.